# Multi-stage bioengineering of a layered oesophagus with in vitro expanded muscle and epithelial adult progenitors

Luca Urbani [1,2], Carlotta Camilli[1], Demetra-Ellie Phylactopoulos[1,3], Claire Crowley[1], Dipa Natarajan[1],
Federico Scottoni [1], Panayiotis Maghsoudlou[1], Conor J. McCann [1], Alessandro Filippo Pellegata [1],
Anna Urciuolo[1], Koichi Deguchi[1], Sahira Khalaf[1], Salvatore Ferdinando Aruta[1], Maria Cristina Signorelli[1],
David Kiely[1], Edward Hannon[1], Matteo Trevisan [1], Rui Rachel Wong[1], Marc Olivier Baradez[4], Dale Moulding[1],
Alex Virasami[5], Asllan Gjinovci[1], Stavros Loukogeorgakis[1], Sara Mantero [6], Nikhil Thapar[1], Neil Sebire[1],
Simon Eaton [1], Mark Lowdell[7], Giulio Cossu[8], Paola Bonfanti [1,3,7] & Paolo De Coppi[1,9]

A tissue engineered oesophagus could overcome limitations associated with oesophageal substitution. Combining decellularized scaffolds with patient-derived cells shows promise for regeneration of tissue defects. In this proof-of-principle study, a two-stage approach for generation of a bio-artificial oesophageal graft addresses some major challenges in organ engineering, namely: (i) development of multi-strata tubular structures, (ii) appropriate re-population/maturation of constructs before transplantation, (iii) cryopreservation of bio-engineered organs and (iv) in vivo pre-vascularization. The graft comprises decellularized rat oesophagus homogeneously re-populated with mesoangioblasts and fibroblasts for the muscle layer. The oesophageal muscle reaches organised maturation after dynamic culture in a bioreactor and functional integration with neural crest stem cells. Grafts are pre-vascularised in vivo in the omentum prior to mucosa reconstitution with expanded epithelial progenitors. Overall, our optimised two-stage approach produces a fully re-populated, structurally organized and pre-vascularized oesophageal substitute, which could become an alternative to current oesophageal substitutes.

[1] Stem Cell and Regenerative Medicine Section, Great Ormond Street Institute of Child Health, University College of London, London WC1N 1EH, UK.
[2] Institute of Hepatology, Foundation for Liver Research, London SE5 9NT, UK. [3] The Francis Crick Institute, London NW1 1AT, UK. [4] Cell and Gene Therapy Catapult, London SE1 9RT, UK. [5] Department of Histopathology, Great Ormond Street Hospital, University College of London, London WC1N 3JH, UK.
[6] Department of Chemistry, Materials and Chemical Engineering "Giulio Natta", Politecnico di Milano 20133, Italy. [7] Royal Free London NHS FT & UCL, London NW3 2QG, UK. [8] Division of Cell Matrix and Regenerative Medicine, Manchester Academic Health Centre, University of Manchester, Manchester M13 9PL, UK. [9] Specialist Neonatal and Paediatric Unit, Great Ormond Street Hospital, London WC1N 1EH, UK. These authors contributed equally: Luca Urbani, Carlotta Camilli. Correspondence and requests for materials should be addressed to P.B. (email: p.bonfanti@ucl.ac.uk)
or to P.D.C. (email: p.decoppi@ucl.ac.uk)

In severe congenital and acquired oesophageal defects, continuity can only be restored by transposing the stomach or gastrointestinal segments into the chest. However, these approaches are complex and associated with serious complications impacting quality of life of recipients[1–6]. Developing functional substitutes for defective oesophagus through combination of biomaterials and patient-derived autologous cells would overcome this unmet clinical need[2–5].

So far, engineered tissues have been successfully applied clinically using decellularized scaffolds to regenerate children's airway[7], and encouraging preclinical data have been obtained for engineering of more complex organs such as gut[8], skeletal muscle[9–11], liver[12,13] and lung[14,15]. Decellularized scaffolds preserve native extracellular-matrix (ECM) overall architecture and composition acting as natural templates guiding cell anchorage, migration, growth and 3D organization in vivo[2–5,16]. Acellular matrices have been previously used as oesophageal substitutes, with successful outcomes only when applied as patches for repairing small defects[17–19] or as tubular devices replacing only mucosa following endoscopic resection[20]. Whole organ regeneration has not yet been achieved since full-thickness circumferential replacements usually lead to strictures[17–19].

The oesophagus is a complex tissue that poses several challenges to clinically successful grafting. First, the oesophagus is multi-layered so requires engineering of all structural compartments for its reconstruction. Transplanting of appropriate cells appears to be key to promote fast, complete and functional regeneration[4,16,21]. In addition, organised and functional scaffold re-population in vitro before transplantation maximizes both the ingrowth of neighbouring host cells and angiogenesis[22–24]. Finally, while previous studies focused on the cervical oesophagus, which is mainly skeletal[17,19–22], thoracic oesophagus is almost exclusively smooth muscle[2–6,16]. Due to these limitations, all previous attempts failed to provide an optimal approach in the use of decellularized scaffolds as suitable oesophageal substitutes[16].

Here, we report for the first time development of a tubular oesophageal ECM engineered via a customized two-step protocol containing both the muscular and epithelial compartments. The use of primary adult precursor cells facilitates the translational impact of the work with smooth muscle, fibroblasts and enteric nervous system (ENS) precursors sequentially combined to build the *muscularis externa*. Epithelial precursors (ROEC) are subsequently seeded creating multi-layered mucosa. Our culture system allows cell engraftment and differentiation on the oesophageal matrix using a newly customised bioreactor enabling functional engraftment in vivo in two transplantation models. Overall our approach provides a fully re-populated, structurally organized and pre-vascularized oesophageal substitute, which could become, in the near future, a novel and valid alternative for treatment of congenital or acquired oesophageal defects.

## Results

**Decellularized rat oesophagi are suitable for cell repopulation.** Rat oesophagi were harvested, luminally cannulated and decellularized via a peristaltic pump. The detergent-enzymatic treatment (DET) produced a pale oesophageal scaffold, which preserved its original size (Fig. 1a). Absence of nuclear material after decellularization was assessed by H&E, DAPI staining, and DNA quantification (Fig. 1a, b). All layers, from lumen to adventitia, were comparable between native and decellularized oesophagi (Fig. 1a, Supplementary Fig. 1a-c). MT also highlighted preservation and correct distribution of collagens (Supplementary Fig. 1a), which were confirmed with collagen type I and IV immunostaining (Supplementary Fig. 1d,e). Comparable amounts

of elastin and glycosaminoglycans (sGAG) were quantified in native oesophagi and decellularized scaffolds (Supplementary Fig. 1f,g). Overall preservation of ECM composition and architecture post-decellularization was reflected in the mechanical properties of the scaffold with no significant difference between native and decellularized oesophageal segments in relaxation, strength and strain at break (Supplementary Fig. 1h).

**Growth of a bio-artificial oesophageal muscle layer.** Primary human mesoangioblasts[25] (hMAB) were expanded up to 10 passages, when they retain a strong proliferation ability (Ki67 expression; Fig. 1c). Expanded hMAB showed typical pericyte marker (alkaline phosphatase [AP], αSMA, NG2 and PDGFRβ) heterogeneity (Fig. 1c). Flow cytometry showed expected variable expression of CD146, CD90, PDGFRβ and AP, almost complete positivity for NG2 and CD44 and no expression of CD45 and CD34 (Supplementary Fig. 2a). The capacity of hMAB to differentiate towards smooth muscle in vitro was assessed by 7 and 14 days exposure to TGFβ1; hMAB acquired typical enlarged smooth muscle cell morphology and were positive for early and mature smooth muscle markers (αSMA, calponin, SM22, and smoothelin; Supplementary Fig. 2b). Mouse fibroblasts (mFB), isolated enzymatically from hindlimb muscles, were expanded in culture and expressed fibroblast markers such as Vimentin and TCF4 (Fig. 1d).

hMAB were initially seeded either on the scaffold surface or delivered by multiple micro-injections. While the latter showed higher cell engraftment, this was still mainly limited to the injection site (Supplementary Fig. 3a-c). In order to improve cell migration within the scaffold, engraftment and distribution were evaluated upon seeding with either hMAB or a combination of hMAB + mFB (85:15). Schematic distribution maps revealed that total cell distribution clearly improved by co-seeding mFB and hMAB (Fig. 1e, f). hMAB alone could engraft into the scaffold with no significant difference in cell number/area compared to hMAB + mFB seeded scaffolds (Fig. 1g). Nevertheless, hMAB-seeded scaffolds were characterised by high heterogeneity and variability in cell distribution, as highlighted by polar distribution (Fig. 1e, f, middle row) and cell density maps (Fig. 1e, f, bottom row). Scaffolds seeded with hMAB + mFB were more homogenous in number of cells per area (Fig. 1f, g) and distribution compared with hMAB-seeded scaffolds (Fig. 1e). Human cells, discriminated from mFB using anti-human Nuclei immunofluorescence staining (Fig. 1h), maintained the initial seeding ratio ie. 23.4 ± 5.06% mFB and 76.6 ± 5.06% hMAB (mean ± SEM) (Fig. 1i).

Co-seeding influenced also the migratory potential of hMAB measured in scaffolds using MTT assay (Fig. 1j). The maximum distance covered by the cells from the centre of the injection point was significantly higher when hMAB were co-seeded with mFB compared to scaffolds seeded with hMAB alone (Fig. 1k). Therefore, co-seeding of both hMAB and mFB was used in the subsequent experiments of this study. When scaffolds were cultured dynamically in a bioreactor (Fig. 2a, b), muscle layer formation and differentiation improved. Dynamic cultured-scaffolds showed significantly higher proliferation of seeded cells and matrix invasion compared to the static culture condition as highlighted by H&E staining and using cell density and polar distribution maps, the latter assuming a perfectly circular section to contrast artefacts (Fig. 2c–e). Using immunostaining for hNuclei and DAPI, hMAB and mFB were identified in randomly selected sections and blindly counted. The percentages of cells at the end of static and dynamic culture were 73.7 ± 4.65% hMAB/ 26.3 ± 4.65% mFB in static and 86.5 ± 4.21% hMAB/13.5 ± 4.21% mFB in dynamic conditions (Fig. 2f). The total cell number and

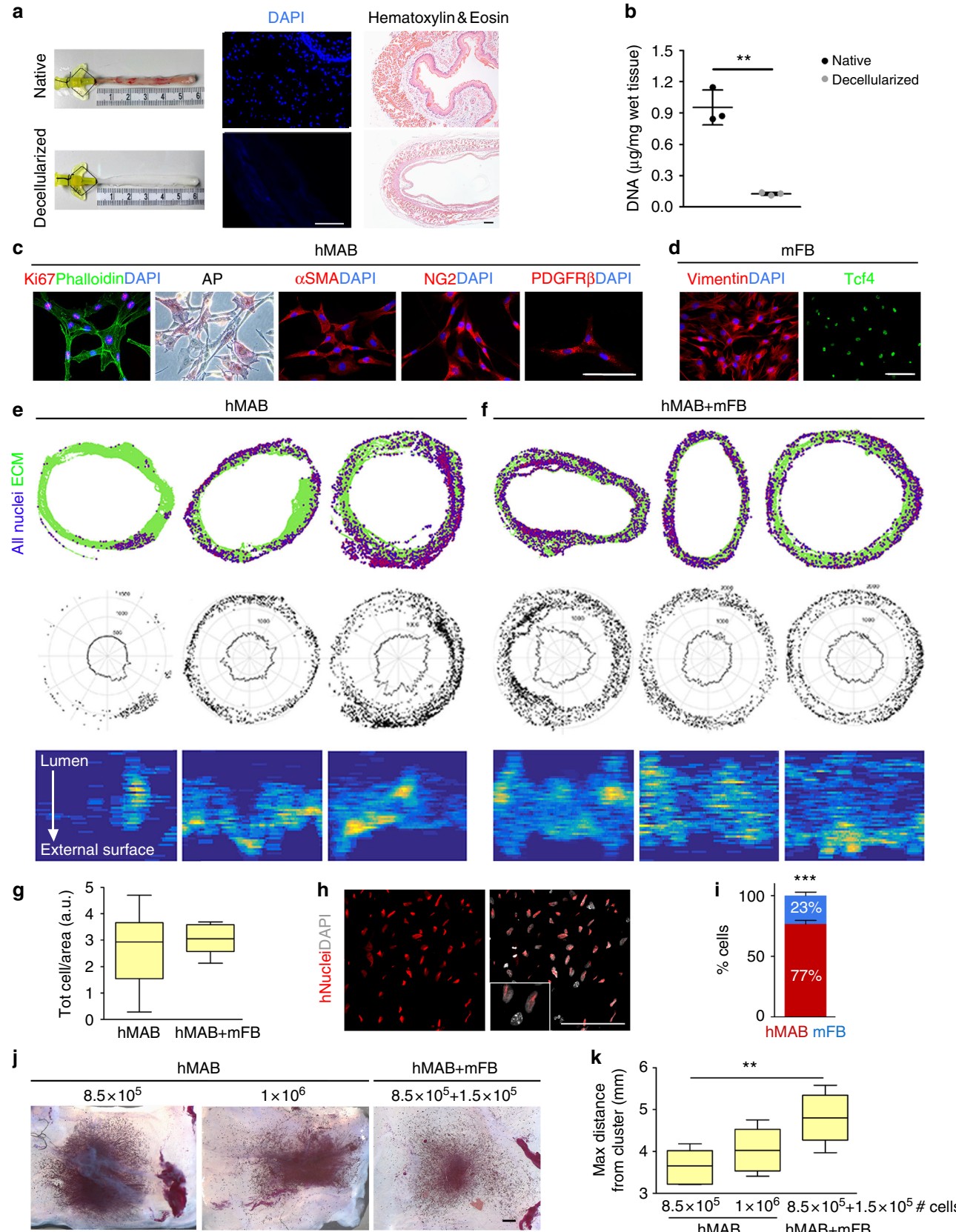

cell density were always significantly higher in scaffolds after dynamic culture compared to static condition (Fig. 2g, h). Seeding, engraftment and viability of hMAB were also monitored with bioluminescence in representative scaffolds using a bioluminescent in vivo imaging system (IVIS). Luc+ZsGreen+hMAB

+ mFB injected in the muscle layer of decellularized scaffolds (day 0) were clearly identifiable in clusters accordingly to the injection sites (Fig. 2i). The average radiance detected in the days following dynamic culture showed an increase in cell number compared to day 0, reaching a stable signal after 11 days of

**Fig. 1** Characterization and re-population of decellularized scaffolds with hMAB and mFB. **a** Macroscopic appearance (left) of native and decellularized rat oesophagus after 2 cycles of DET. DAPI staining (centre) of native and decellularized scaffold sections to identify nuclei. Scale bar: 100 μm. Hematoxylin and eosin (right) of sections of native oesophagus and decellularized scaffold. Scale bar: 100 μm. **b** DNA quantification in samples of native and decellularized oesophagi. Data: mean ± SD ($n = 3$; **$p = 0.0011$; $t$-test). **c** Characterization of hMAB isolated from paediatric skeletal muscle biopsies and expanded for up to 10 passages in culture: immunofluorescence staining for Ki67 and phalloidin, colorimetric staining for alkaline phosphatase, immunofluorescence staining for αSMA, NG2 and PDGFRβ. Nuclei were stained with DAPI. Scale bar: 100 μm. **d** Characterization of mFB isolated from murine skeletal muscles and expanded for up to 7 passages in culture: immunofluorescence staining for vimentin and Tcf4. Nuclei were stained with DAPI. Scale bar: 100 μm. **e**, **f** Top row: cell distribution maps showing single cells (purple) and ECM scaffold (green) in sections of 3 representative scaffolds seeded with hMAB only (**e**) or hMAB + mFB (**f**) and cultured for 6 to 9 days in static conditions. Cells were identified using DAPI staining. Middle row: polar distribution maps obtained from the correspondent cell distribution maps (in the top row) assuming a perfectly circular section of the scaffolds. Bottom row: cell density maps obtained from the maps above showing cell density in seeded scaffolds from the lumen (top) to the external surface (bottom). **g** Total number of cells per area in scaffolds seeded with hMAB only or hMAB + mFB and cultured for 6 to 9 days expressed as 25th to 75th percentile, median and min to max whiskers ($n = 6$). **h** Immunostaining for hNuclei and DAPI used to discriminate between hMAB and mFB. Scale bar: 100 μm. **i** Proportion of hMAB and mFB in co-seeded scaffolds after culture. Data: mean ± SEM ($n = 3$; ***$p = 0.0002$ hMAB vs mFB; $t$-test). **j** Representative images of MTT colorimetric assay on scaffolds seeded with $8.5 \times 10^5$ hMAB or $1 \times 10^6$ hMAB with/without mFB (ratio 85:15) and cultured in static for 6 days. Scale bar: 1 mm. Viable cells are coloured in purple. **k** Graph shows the measure of the maximum distance in mm covered by the cells from the centre of the cell cluster to the empty edge of the scaffolds. Data: 25th to 75th percentile, median and min to max whiskers ($n = 6$; **$p = 0.0049$; ANOVA)

culture (Fig. 2j). Most importantly, cell distribution along the scaffold clearly improved immediately after cell seeding, providing cellular coverage of the scaffold and a resultant homogenous cell distribution. Interestingly, the bio-engineered muscle could be cryopreserved with a slow-cooling process showing maintenance of cell viability after storage. Oesophageal scaffolds seeded with Luc+ZsGreen+hMAB + mFB were cultured in static conditions and then cryopreserved for 2 weeks. Post-thawing, scaffolds showed a slight reduction in cell viability when compared to before cryopreservation. However, cells were able to recover and grow for up to 7 days in static culture, as confirmed by bioluminescence reading and MTT assay (Fig. 2k–m).

The dynamic culture also significantly improved smooth muscle differentiation of hMAB (Fig. 3). Schematic cell distribution maps were obtained from SM22 and hNuclei co-staining of scaffolds seeded with hMAB + mFB and cultured in either static or dynamic conditions. In static culture, a few layers of SM22+ cells were located mainly on the external surface of the scaffold even after prolonged culture in differentiation medium (Fig. 3a). The dynamic culture resulted in an overall homogenous distribution of smooth muscle differentiated hMAB in all the layers of the scaffold (Fig. 3b). Immunofluorescence highlighted how cells were oriented alongside the ECM of the scaffold, resembling the dual-layered structure of native oesophageal muscle (Fig. 3b). In particular, SEM showed how cells migrated through the matrix pores, which were evident in the unseeded scaffolds, scattered among the decellularized muscle fibres (Fig. 3c). More extensive smooth muscle differentiation was confirmed by cell counting, showing a significant higher percentage of SM22+ cells in dynamic-cultured scaffolds compared to static conditions (Fig. 3d). Interestingly, both culture conditions showed a comparable contribution of mFB to the total amount of SM22+ cells. Mature smooth muscle differentiation was confirmed by positivity of hNuclei+ cells for early and late-stage differentiation markers, namely αSMA, calponin and connexin43 (Fig. 3e–g). The pattern of calponin and SM22 co-expression was comparable to native human oesophagi (Fig. 3f).

Differentiation was also evaluated by investigating oxidative metabolism ($^{13}CO_2$ production from $^{13}C$-glucose) in 2D cultures. The increase in oxidative metabolism was not triggered by the co-presence of mFB but correlated to hMAB smooth muscle differentiation by addition of TGFβ1 (Supplementary Fig. 4). Dynamic 3D culture conditions further increased hMAB oxidative metabolism compared to static conditions (Fig. 3h).

Beside differentiation, dynamic conditions also promoted proliferation of hMAB (Fig. 3i, j) with very few apoptotic caspase3+ cells after 11 days of 3D culture (Fig. 3k).

**Enteric neural cells functionally engraft in the muscle layer.** YFP-expressing murine neural crest cells (mNCC) were isolated from postnatal Wnt1-cre;Rosa26$^{YFP/YFP}$ guts. mNCC formed neurospheres in culture, and expressed markers of enteric neural stem cells (Sox10) and derivatives, namely glia (S100) and neurons (TuJ1) (Fig. 4a). mNCC engrafted in both static and dynamic cultures along with hMAB and mFB (Fig. 4b–e). Although the number of GFP+mNCC was significantly lower after dynamic culture (Fig. 4f), mNCC spreading from the injection site was more evident in dynamic-cultured scaffolds. mNCC distributed throughout the scaffold, proliferated extensively and arranged themselves in ring-like structures similar to those observed in native oesophagi (Fig. 4b–e). mNCC differentiated readily in both static and dynamic cultures into both neurons and glia as seen by immunostaining for TuJ1 and S100, respectively, and displayed formation of connections with hMAB (Fig. 4g–k). In addition, both evoked and post stimulation spontaneous calcium transients were observed in YFP+mNCC-derived cells using the calcium indicator Rhod-3 and electrical point stimulation of YFP+ nerve fibres present in dynamic-cultured scaffolds (Fig. 4l, m). Remarkably, a representative muscle layer engineered with hMAB + mFB + mNCC in dynamic conditions for 11 days showed schematic cell distribution, cell density and number of cells per area comparable to the muscle layer of a native rat oesophagus (Supplementary Fig. 5a-c).

**ROEC repopulate the inner surface of decellularized scaffolds.** Rat oesophageal epithelial cells (ROEC) were isolated from adult rats and plated over lethally-irradiated feeder layers for expansion. ROEC gave rise to round epithelial colonies (Fig. 5a) and were sub-cultured weekly for at least two months with colony forming efficiency of 10–15% (Fig. 5b). Independent cultures were obtained consistently from different animals ($n = 6$). In the first 3 days ROEC grew slowly, then exponentially with a doubling time of 18–19 h (Fig. 5c). Cytokeratin (CK)5/14 was expressed by all cells in growing colonies; however, with colony size increasing over a week, ROEC in the colony centre would start differentiating, thus expressing CK13 (negative in early cultures), as in the uppermost layers of oesophageal epithelium (Fig. 5d; Supplementary Fig. 6a). Colonies consisted of highly proliferative epithelial cells, E-Cadherin+/Ki67+ (Fig. 5e) and

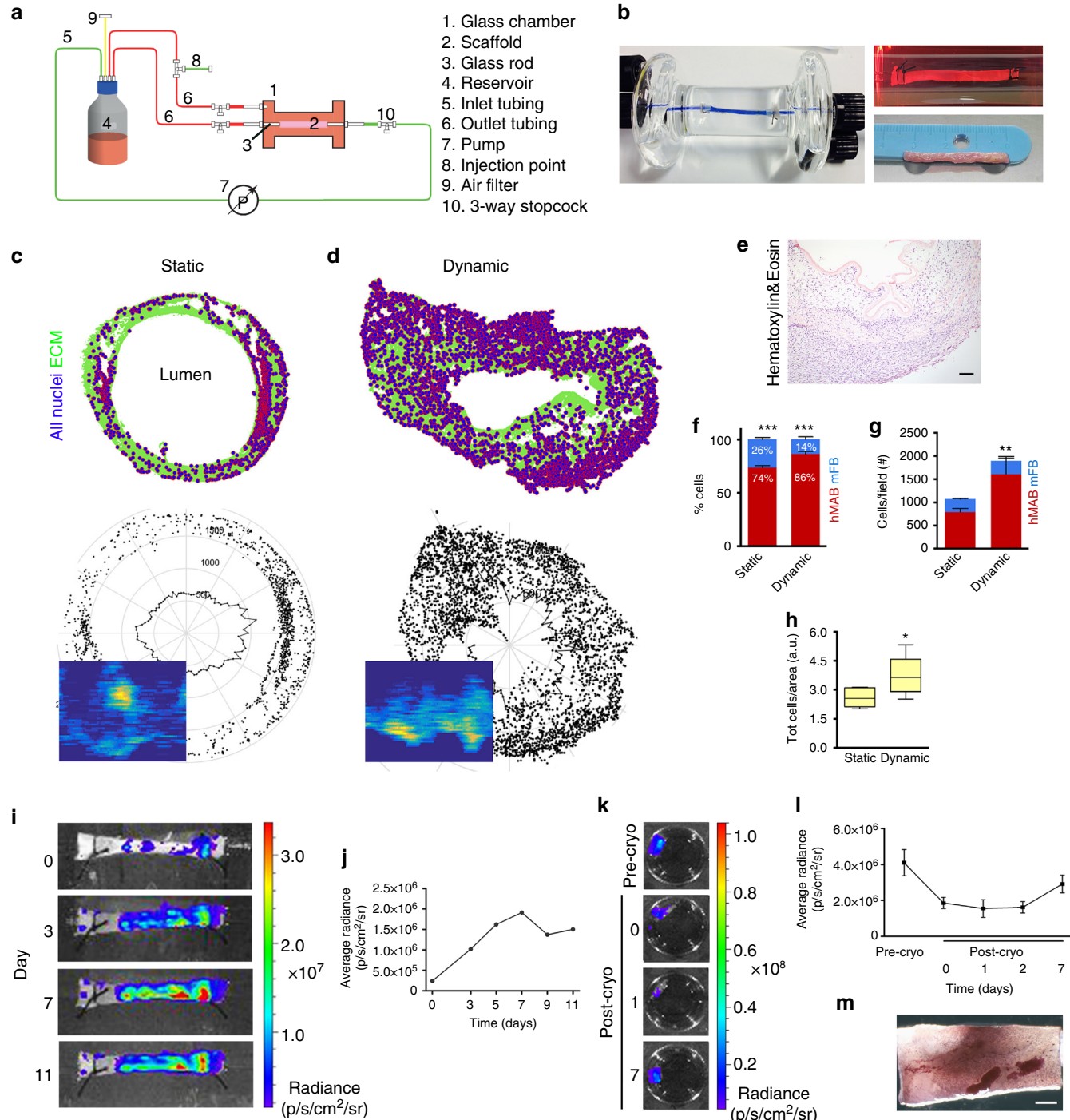

expressed also p63 (Fig. 5f). ROEC were sub-cultured every 6 to 7 days when they reached a sub-confluent density.

After in vitro expansion ROEC were seeded on the luminal surface of decellularized scaffolds where they distributed along the entire surface (~1 cm²) and in 3 days formed a highly proliferative monolayer expressing the basal marker CK5/14 (Fig. 5g; Supplementary Fig. 6b). Importantly, ROEC gave rise to a multi-layered epithelium after 2 weeks in vitro (Fig. 5h; Supplementary Fig. 6c). Upon stratification, Ki67+ proliferating cells were detected only in the basal layer (Fig. 5h), whereas the uppermost layers had downregulated CK5/14 and expressed CK13, E-cadherin and tight junction protein (ZO-1), as in the native oesophagus (Fig. 5h, i; Supplementary Fig. 6d,e). In order to evaluate the ability of epithelial cells to engraft, polarize and migrate in a tubular scaffold, Luc+ZsGreen+ROEC were seeded directly within the bioreactor. Bioluminescence images detected transduced cells along the entire length of the scaffold after seeding (day 0), as well as after the introduction of the flow and until the end of the experiment (day 1 to 3) (Fig. 5j). Photon emission analysis of the transduced cells was also performed (Fig. 5k): at day 1, there was a slight decrease in photon emission, probably due to removal of non-adherent floating cells, whereas at day 2 cells reached a peak, demonstrating that engrafted cells were viable, and proliferative (Fig. 5k). Histological analysis demonstrated that ROEC grew throughout the luminal surface of the decellularized scaffold forming a homogeneous monolayer

**Fig. 2** Decellularized scaffolds seeded with hMAB + mFB: static vs. dynamic culture and cryopreservation. **a** Simplified schematic of the bioreactor used for dynamic culture. **b** Photo of the glass chamber used for dynamic culture showing a decellularized oesophageal scaffold sutured between the 2 glass rods (left). Blue dye was used to highlight separation between the luminal and external compartments. Picture of a representative scaffold, seeded with hMAB + mFB, during culture in the glass chamber in dynamic conditions (top right) and at the end of the culture (bottom right). **c, d** Cell distribution maps, polar distribution maps and cell density maps obtained from DAPI-stained sections of a representative scaffold seeded with hMAB + mFB and cultured in static (**c**) or dynamic (**d**) conditions for 11 days. **e** Hematoxylin and eosin staining of a representative section of a decellularized scaffold seeded with hMAB + mFB and cultured in dynamic conditions for 11 days. Scale bar: 100 μm. **f** Proportion of hMAB and mFB in co-seeded scaffolds after 11 days of culture in static or dynamic conditions. Data: mean ± SEM ($n = 3$-$6$; ***$p < 0.0001$ hMAB vs mFB; $t$-test). **g** Total number of cells per field identified in sections of recellularized scaffolds stained for hNuclei and DAPI, with proportion of hMAB and mFB in static and dynamic-cultured scaffolds. Data: mean ± SEM ($n = 6$-$11$; technical replicates; **$p = 0.0077$ static vs dynamic; 2-way ANOVA). **h** Total number of cells per area in scaffolds recellularized with hMAB + mFB for 11 days expressed as 25th to 75th percentile, median and min to max whiskers ($n = 6$-$14$; technical replicates; *$p = 0.0116$; $t$-test). **i** Bioluminescence images of a representative scaffold seeded with Luc$^+$ZsGreen$^+$hMAB + mFB and cultured in dynamic conditions for 11 days. **j** Graph of the average radiance measured at different time points. **k** Bioluminescence images of a representative scaffold seeded with Luc$^+$ZsGreen$^+$hMAB + mFB and cultured in static for 8 days (pre-cryo) and for further 7 days after 2 weeks of cryopreservation. **l** Average radiance detected before and after cryopreservation at different time points. Data: mean ± SEM ($n = 3$). **m** Representative image of MTT colorimetric assay performed on a seeded-scaffold following cryopreservation. Scale bar: 1 mm

(Fig. 5l) that immunostaining confirmed to be a proliferative basal layer positive for CK5/14, p63 and Ki67 (Fig. 5m–p).

**Two-stage oesophageal muscle-epithelium engineering**. Scaffolds seeded with hMAB + mFB + mNCC and cultured in dynamic conditions were implanted in the omentum or underneath the kidney capsule of NOD-SCID-gamma (NSG) mice to assess if different in vivo environments could affect graft survival and vascularisation. Omental implantation was followed by seeding of ROEC to re-establish the epithelium (Fig. 6a).

Engineered muscle layers implanted underneath the kidney capsule showed the presence of numerous SM22$^+$hMAB 7 days post implantation (Supplementary Fig. 7a-f). Similarly, seven days post omental implantation, hMAB showed preserved localization and smooth muscle phenotype, indicated by hNuclei$^+$ cells co-stained for SM22 (Fig. 6b, c). Importantly, cells still aligned with the pre-existing ECM structure of the muscle layer, expressing mature smooth muscle markers such as connexin43 and calponin (Fig. 6d, e). mNCC also showed morphology and distribution comparable to that observed in pre-implantation dynamic-cultured scaffolds, with cross-muscle projections and widespread distribution (Fig. 6e, f). A subset of hMAB maintained their proliferative capability in vivo ($10.0 ± 1.55\%$ Ki67$^+$ cells were detected inside the oesophageal scaffold, of which $72.6 ± 21.01\%$ were hNuclei$^+$ positive – data not shown), with a very low rate of apoptosis ($1.5 ± 0.36\%$ caspase3$^+$ cells) (Fig. 6g, h). Angiogenesis was evident from the omentum towards the external muscle layer of the implanted grafts. These neo-capillaries were vWF-positive and, although surrounded by hNuclei$^+$ cells, no hNuclei$^+$vWF$^+$ cells were found (Fig. 6i). No migration of hMAB or YFP$^+$mNCC was detected in tissues surrounding implanted seeded scaffolds. A certain degree of expected inflammatory reaction was detected 1 week post-implantation, with infiltration of F4/80$^+$ macrophages and Ly6G$^+$ neutrophils in the *muscularis externa* of the scaffold (Supplementary Fig. 7g). Control scaffolds showed very limited host cell invasion of the matrix and reduced neo-vascularization (Fig. 6j), with no hNuclei or GFP staining.

Dynamic-cultured scaffolds seeded with hMAB + mFB + mNCC were harvested 1 week after in vivo omental implantation and seeded with ROEC. This two-stage seeding approach allowed in vitro and in vivo smooth muscle maturation, graft neo-vascularization and then epithelial cell engraftment. ROEC were seeded luminally and formed a monolayer with highly proliferative E-cadherin, CK14, p63, and PanCytokeratin positive cells (Fig. 6k–n). After one week, cells started to differentiate as demonstrated by CK13 expression (Fig. 6l). Caspase3 staining

identified a few apoptotic cells, showing that most seeded cells were still viable (Fig. 6o).

## Discussion

Here we describe a novel engineering of a morphologically and functionally organised oesophagus using a step-by-step seeding of primary cells, capable of proper assembly within a decellularized scaffold and efficient differentiation in a newly customised bioreactor. Importantly, this engineered oesophagus can be cryopreserved, is able to engraft and becomes vascularized when transplanted in vivo.

Decellularized oesophagus was obtained by adapting a previously reported technique optimized for simple tissues (skin, skeletal muscle); tubular structures (trachea, intestine); or more complex tissues (liver, lung, kidney)[7,26–29]. DET removed cellular components, avoiding antigenicity reaction, but preserved the major ECM molecules, maintaining elastin and sGAG content, distribution of collagen I and IV, laminin, and the overall multi-strata architecture. These characteristics assured biomechanical performances such as strength, distensibility and stiffness of decellularized oesophagi comparable with the ones of native tissues, as evaluated with several mechanical tests, and consistent with other studies using DET for tubular organs[29–31].

Avoiding cadaveric derived scaffolds (either of human or animal origin) and using synthetic polymers would have the advantage of having an off-shelf product and eliminate the potential risks of infections and organ shortage[32]. The synthetic scaffolds could be designed to recapitulate the various oesophageal layers and preloaded with specific growth factor capable to both allow cell proliferation and differentiation. This smart manufacturing, which may even avoid the need for cell seeding prior to transplantation making it cheaper and safer for patients is, however, still limited to small scale[33]. Avoiding cell seeding may also be possible in case the polymer is used as a stent to drive oesophageal regeneration but it is ultimately removed endoscopically[34], or for the repair of small oesophageal defects which do not affect the all oesophageal circumference[35]. Orthotopic implantation of unseeded decellularized scaffolds leads to stenosis, strictures and lack of function with poor or mixed clinical outcome[17,19,36–38], nevertheless the positive impact of ECM from decellularized tissues on oesophageal healing process have been reported in canine and porcine models[38,39]. Promising results have been shown also when unseeded decellularized matrices were used for oesophageal reconstruction in humans. Full-thickness patch repairs and circumferential mucosal substitutions have been attempted with notable outcomes[40,41]. More recently,

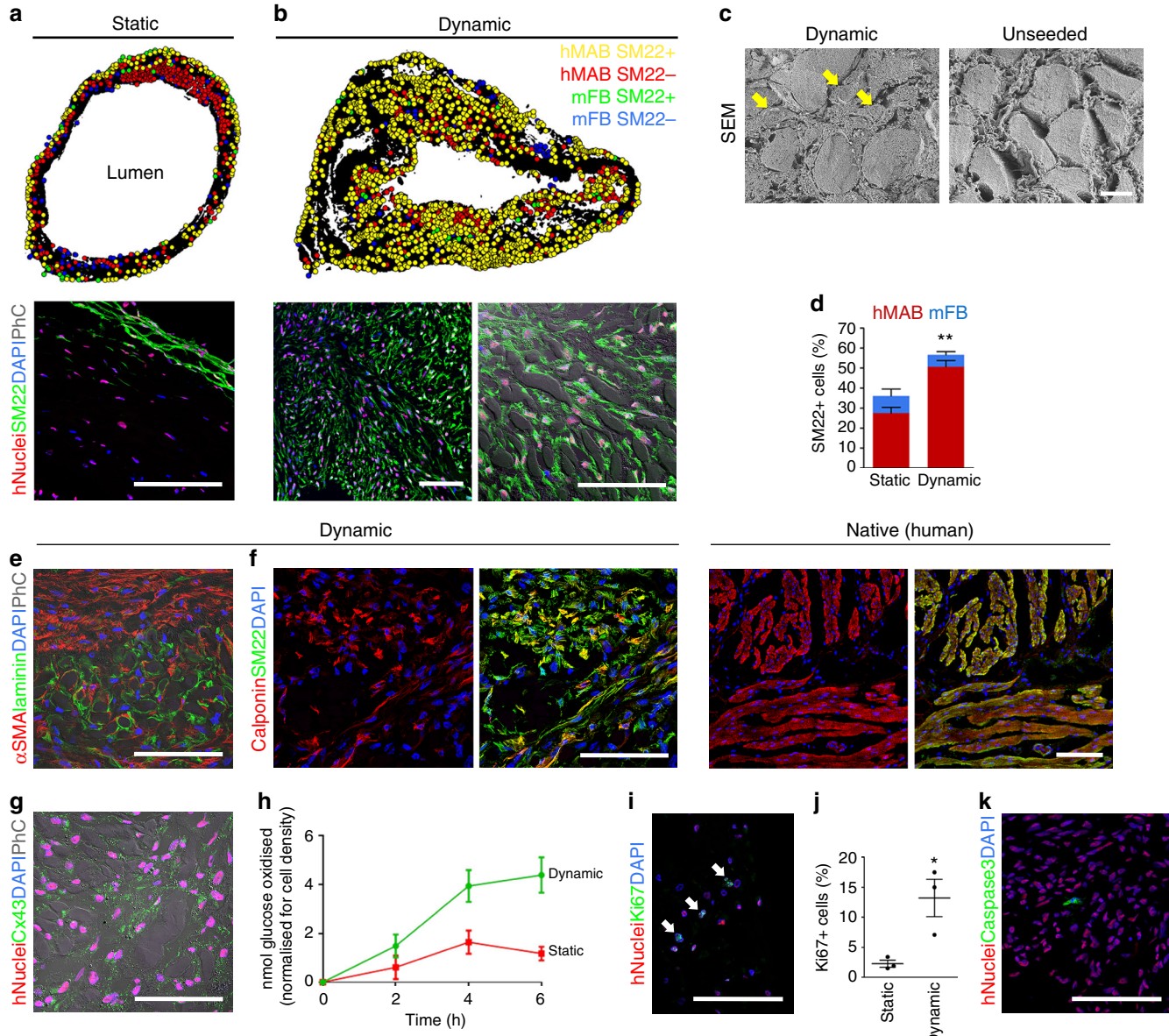

**Fig. 3** Smooth muscle differentiation and proliferation in static and dynamic cultured re-cellularized scaffolds. **a, b** Top: cell distribution maps obtained from sections stained for hNuclei, SM22 and DAPI, of representative scaffolds seeded with hMAB + mFB and cultured in static (**a**) or dynamic (**b**) culture for 11 days. Bottom: representative images of immunofluorescence for hNuclei, SM22, DAPI (with phase contrast). Scale bar: 100 μm. **c** Scanning electron microscope images of dynamic-cultured (left) and unseeded scaffolds (right). Arrows indicate seeded cells among the pre-existing muscle fibres. Scale bar: 10 μm. **d** Percentage of SM22$^+$ cells per field identified in sections of recellularized scaffolds with proportions of hMAB and mFB positive for SM22, in static and dynamic-cultured scaffolds. Data: mean ± SEM ($n = 3$; **p = 0.0024 static vs dynamic; $t$-test). **e–g** Representative images of immunofluorescence for αSMA and laminin (with phase contrast) (**e**), calponin and SM22 (**f**), hNuclei and connexin43 (with phase contrast) (**g**), in sections of scaffolds seeded with hMAB + mFB and cultured in dynamic conditions for 11 days. Native human oesophagus was used as comparison (**f**). Nuclei were stained with DAPI. Scale bar: 100 μm. **h** Oxidative metabolism ($^{13}$C-glucose oxidation assay) measured as $^{13}CO_2$ production in cultured media sampled every 2 h at the end of the static and dynamic cultures (11 days). $^{13}CO_2$ quantification (in nmol) was normalised for the total number of cells in the corresponding scaffolds. Data: mean ± SEM ($n = 3$; technical replicates). **i** Representative image of immunofluorescence for hNuclei, Ki67 and DAPI in a section of scaffold recellularized with hMAB + mFB after 11 days of dynamic culture. Scale bar: 100 μm. **j** Percentage of Ki67$^+$ cells per field identified in sections of recellularized scaffolds cultured in static and dynamic conditions. Data: mean ± SEM ($n = 3$; *p = 0.0265; $t$-test). **k** Representative image of immunofluorescence for hNuclei, caspase3 and DAPI of a section of scaffold recellularized with hMAB + mFB after 11 days of dynamic culture. Scale bar: 100 μm

an unseeded biological and synthetic combined scaffold was successfully used for bridging a circumferential full-thickness defect in 1 patient[35]. Seeding of cells prior to implantation led to more viable scaffolds with less pronounced inflammatory response and increased ingrowth of epithelial and smooth muscle cells[20,22–24]. Previous studies combined non-tissue-specific ECM with epithelial cells, muscle cells or adipose tissue-derived stem

cells[22,23,37,38]. Luc and colleagues tested different strategies for scaffold re-population with adipose tissue-derived stem cells, obtaining cell adhesion and migration to the external and luminal side of decellularized porcine oesophagi[38]. Upon transplantation in nude rats, re-cellularized scaffolds showed host cells invasion, vascularization and strong ECM remodelling/absorption. However, the contribution by seeded stem cells to these in vivo

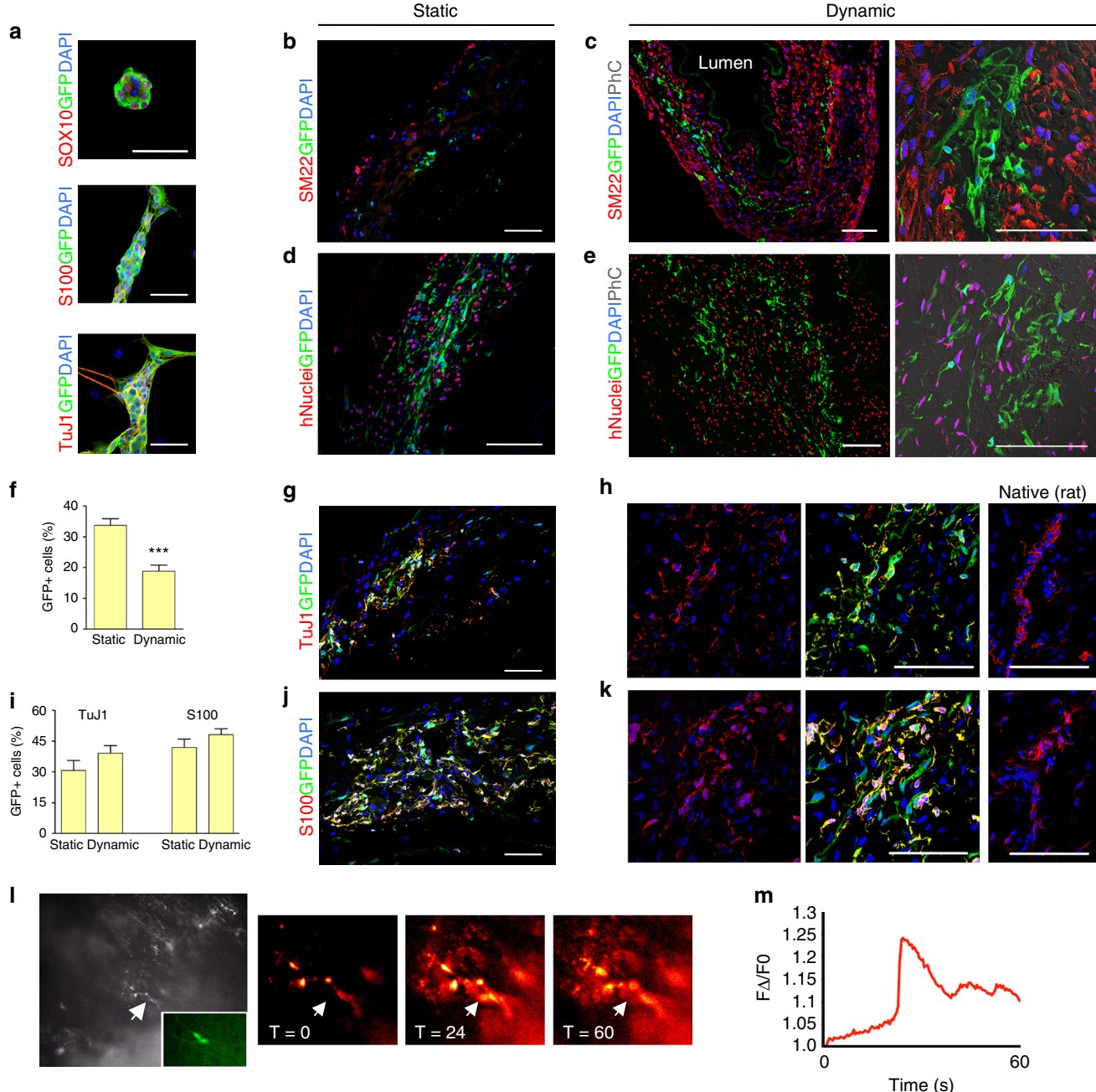

**Fig. 4** Characterization of mNCC and co-seeding with hMAB and mFB in decellularized scaffolds. **a** Characterization of mNCC isolated from Wnt1-cre; Rosa26$^{YFP/YFP}$ murine gut and expanded in culture as neurospheres. Immunofluorescence staining for SOX10, S100 and TuJ1 and epifluorescence for GFP. Nuclei were stained with DAPI. Scale bar: 50 μm. **b**–**e** Representative images of immunofluorescence for SM22, GFP and DAPI (**b**, **c**) and hNuclei, GFP and DAPI (**d**, **e**) (with phase contrast - PhC) in sections of scaffolds seeded with hMAB + mFB + mNCC and cultured in static and dynamic conditions for 11 days. Scale bar: 100 μm. **f** Percentage of GFP$^+$ cells per field identified in sections of static and dynamic-cultured scaffolds. Data: mean ± SEM ($n = 12$–16; technical replicates; ***$p < 0.0001$ static vs dynamic; $t$-test). **g**–**k** Representative images of immunofluorescence for S100 and GFP or TuJ1 and GFP in sections of static and dynamic-cultured scaffolds. Native rat oesophagus was used as comparison (right). Nuclei were stained with DAPI. Scale bar: 100 μm. **i** Percentage of TuJ1$^+$GFP$^+$ and S100$^+$GFP$^+$ cells per field identified in sections of static and dynamic-cultured scaffolds. Data: mean ± SEM ($n = 12$–16; technical replicates). **l** Epifluorescence in black and white and in green (inlet) (left) and Ca$^{2+}$ imaging at different time points (sec) (right) of a mNCC cell (arrow) identified on the surface of a representative scaffold after 11 days of dynamic culture. **m** Calcium transient plot

processes was unclear and only partially addressed. So far, no study has shown re-population of multiple layers with a combination of different progenitor types prior to implantation.

hMAB, fibroblasts and mNCC were seeded in decellularized scaffolds via multiple micro-injections. This technique provided better cell distribution and engraftment in all layers of the muscle, an essential prerequisite to complete scaffold re-population,

without affecting its integrity. In the human oesophagus, most of the *muscularis externa* is composed of smooth muscle whereas striated muscle predominates in the upper third. Published studies on seeding of muscle cells in decellularized scaffolds have been focused on smooth muscle cells or mesenchymal stem cells[3,20,23,37]. Seeding smooth muscle cells was associated with less inflammatory reaction, enhanced muscle regeneration and

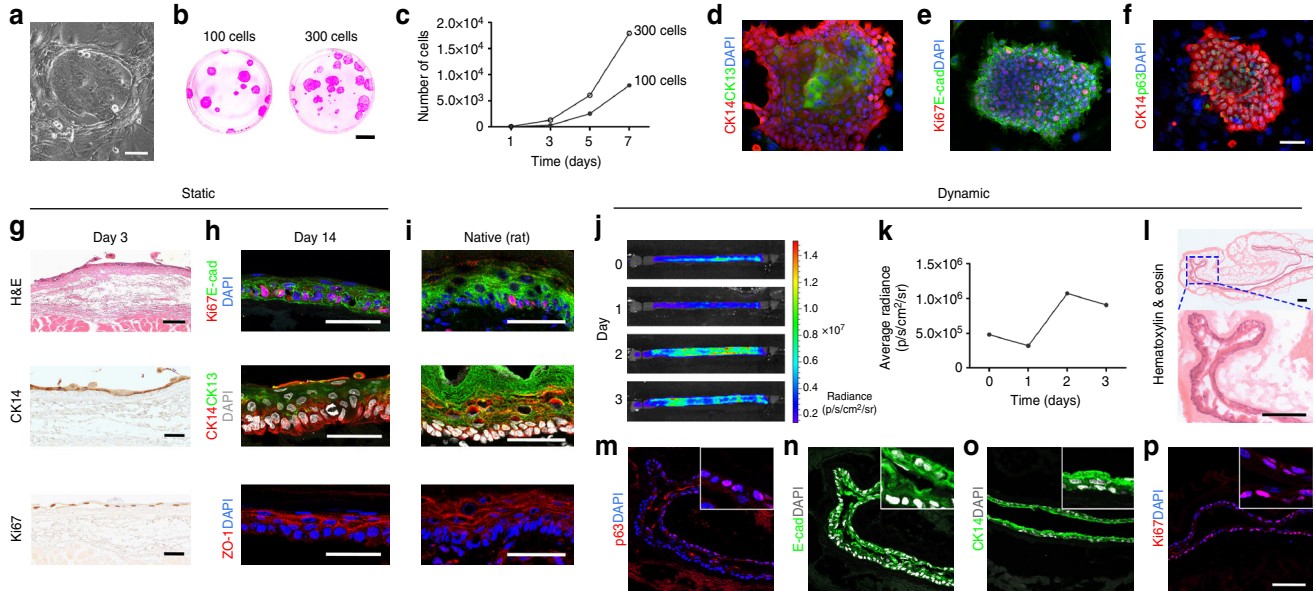

**Fig. 5** Characterization and seeding of ROEC in decellularized scaffolds. **a** ROEC growing over irradiated feeder layer. Scale bar: 100 μm. **b** Colony forming efficiency of 100 and 300 ROEC stained with Rhodamine B. Scale bar: 1.5 cm. **c** Growth rate of representative ROEC as total number of cells at different days of culture. **d–f** Immunofluorescence of ROEC for the expression of CK14 and CK13 (**d**), Ki67 and E-cadherin (**e**) and CK14 and p63 (**f**). Nuclei were counterstained with DAPI. Scale bar: 66 μm. **g** Hematoxylin and eosin (H&E) staining, immunostaining for CK14 and Ki67 in sections of scaffolds seeded with ROEC and cultured in static conditions for 3 days. Scale bar: 100 μm. **h** Immunofluorescence staining for Ki67 and E-cadherin, CK14 and CK13, and ZO-1 after 14 days of culture. Nuclei were counterstained with DAPI. Scale bar: 50 μm. **i** Immunofluorescence staining of a native rat oesophagus for Ki67 and E-cadherin, CK14, CK13 and ZO-1. Nuclei were counterstained with DAPI. Scale bar: 50 μm. **j** Bioluminescence images of a representative scaffold seeded with Luc+ZsGreen+ROEC and cultured in dynamic conditions for 3 days. **k** Graph of the average radiance measured from the whole scaffold at different time points. **l** Hematoxylin and eosin staining in a section of scaffold seeded with ROEC and cultured in dynamic conditions for 3 days. Scale bar: 100 μm. **m–p** Immunostaining for p63 (**m**), E-cadherin (**n**), CK14 (**o**) and Ki67 (**p**) in sections of scaffolds seeded with ROEC and cultured in dynamic conditions for 3 days. Nuclei were counterstained with DAPI. Insets show staining specificity and localization. Scale bar: 100 μm

lower risk of strictures in vivo when compared to unseeded scaffolds[23]. However, smooth muscle cells are difficult to expand while mesoangioblasts isolated from skeletal muscle can easily proliferate in culture and are able to undergo both spontaneous skeletal and TGF-beta-induced smooth muscle differentiation[42–44]. hMAB are also referred to as pericyte-derived cells since they express alkaline phosphatase, NG2 proteoglycan and PDGFRβ[25,45]. They have been successfully used in combination with synthetic scaffolds to promote regeneration of vascular grafts and skeletal muscle[46,47] and, importantly, they have already been transplanted in patients[48]. However this is the first study reporting smooth muscle regeneration by combining mesoangioblasts and decellularized scaffolds. hMAB engrafted in the decellularized oesophageal *muscularis externa*, but their homogeneous distribution improved by co-seeding murine fibroblasts in a dynamic culture. Fibroblasts promote local stem cell recruitment and secrete ECM proteins, such as collagen VI and fibronectin, and trophic factors during tissue regeneration[49–51]. Although they have potential to support scaffold seeding[9,50], little has been reported on their role in the generation of artificial tissues. Fibroblasts promoted hMAB scaffold invasion, but did not increase cell engraftment, highlighting that their effect could be related to ECM remodelling and secretion of pro-migratory cytokines rather than proliferative ones. Co-culturing of hMAB, mFB and mNCC for 11 days in dynamic conditions produced a construct with cell density and distribution features similar to the native counterpart. This supports our hypothesis that the combination of hMAB with other cell types fabricates a multi-layered tissue that can grow and specialize in vitro. The widespread hMAB distribution was maintained during differentiation into mature smooth muscle; cells displayed a homogenous parallel cell

orientation and expressed smooth muscle specific markers. Importantly, hMAB filled pores created by decellularization and aligned with pre-existing fibres, which may play a role in differentiation, as described for other muscle cells[52]. Moreover, dynamic culture provided intraluminal pressure and improved diffusion of media and growth factors maximising cell growth and differentiation, as observed for engineered vascular grafts. In static cultures, only hMAB on the external surface of the scaffold differentiated towards smooth muscle, with minimal change in their metabolic profile compared to dynamic conditions where cells showed more oxidative metabolism[53]. Interestingly, a fraction of hMAB preserved the capacity to proliferate, as demonstrated by the presence of Ki67+ hMAB after 11 days of culture, which is essential for tissue regeneration after transplantation. Muscle functionality however requires enteric nervous system (ENS), which is important for peristaltic motor patterns, and for secretion of enzymes, hormones and neuropeptides. NCC are precursors of the ENS components and their culture conditions allow maintenance of both undifferentiated and more differentiated neurons and glial cells[54]. Therefore, when injected into the scaffold, we observed that, despite being maintained in myogenic media, undifferentiated mNCC increased in number; the mature neurons and glial cells made connections with muscle cells and, most importantly, mNCC were seen to be arranged in two concentric rings. This result, together with preliminary calcium imaging studies, suggested that with relatively short culture periods of 11 days, functional electrical connections are established within the scaffold providing evidence of a rudimentary circuitry.

In order to reconstitute the barrier function of tissue-engineered oesophagus, we required epithelial cells that can

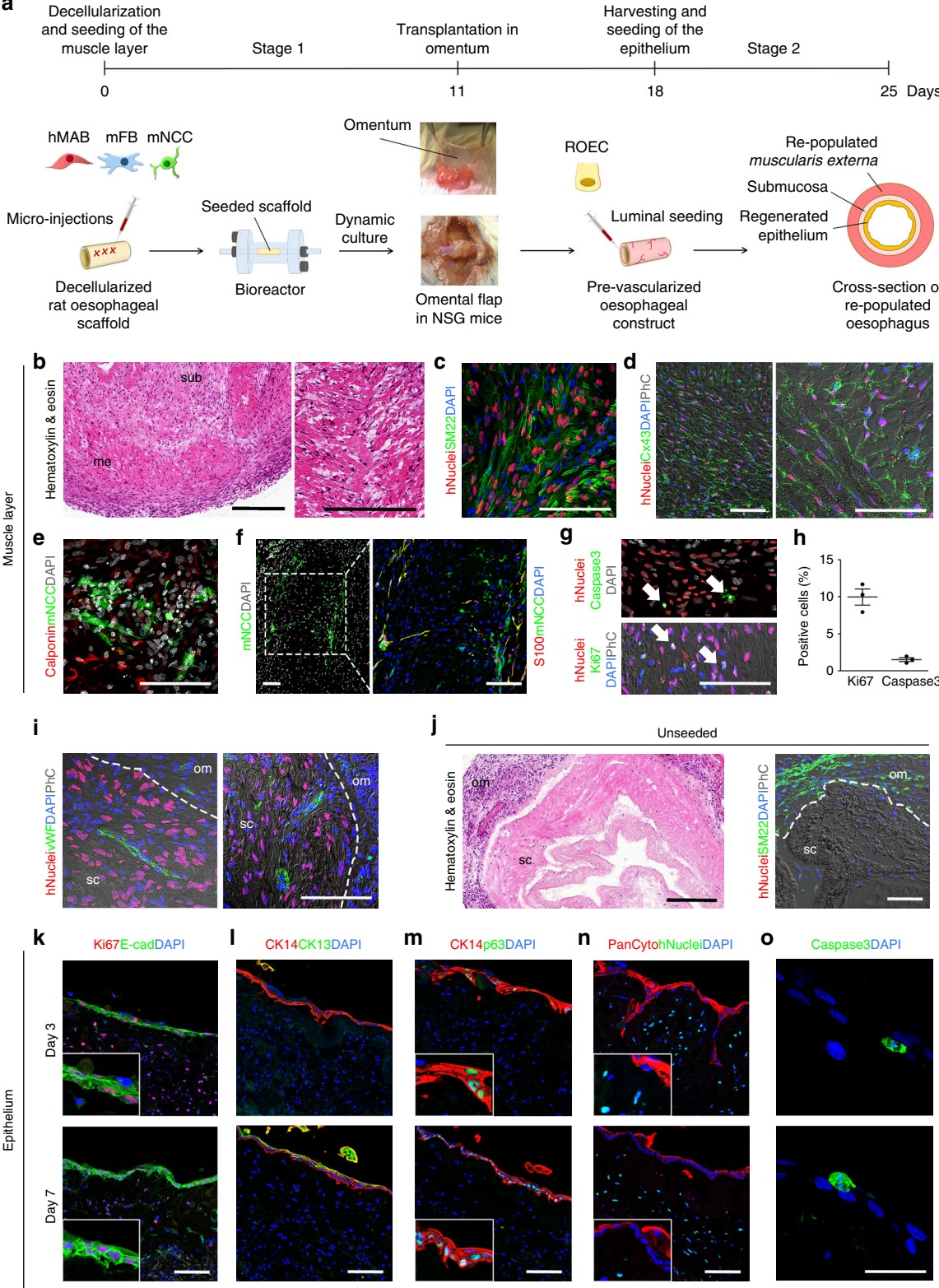

expand extensively in vitro while maintaining capacity to differentiate and stratify on the decellularized scaffold. ROEC were isolated and grown from single cells in co-culture with feeders. This method was first described for human keratinocytes and then for other stratified epithelia that have reached clinical application and are now consolidated therapies[55,56] ROEC were expanded long-term giving rise to colonies constituted by cells

with features of a basal layer (e.g. Cytokeratin 5/14$^+$, Ki67$^+$ and p63$^+$). Colony-forming efficiency assay demonstrated that a proliferating pool of cells was consistently carried over in subsequent passages. Following expansion, ROEC seeded on the luminal surface of scaffolds were able to properly differentiate by forming a multi-layered epithelium with a clearly defined CK5$^+$ basal layer and differentiated CK5$^-$CK13$^+$ suprabasal layers that

**Fig. 6** Two-stage re-population process: muscle maturation, pre-vascularization in vivo and restoration of epithelium. **a** Schematic of the two-stage re-population process. **b–h** Analysis of scaffolds seeded with hMAB + mFB + mNCC, cultured in dynamic conditions for 11 days, implanted in the omentum of NSG mice and harvested after 7 days. All images show the *muscularis externa* of the re-populated scaffolds. **b** Hematoxylin and eosin staining. Sub: submucosa; me: *muscularis externa*. Scale bar: 250 μm. **c** Immunofluorescence staining for hNuclei, SM22 and DAPI. Scale bar: 100 μm. **d** Immunofluorescence staining for hNuclei, Connexin43 and DAPI (with phase contrast). Scale bar: 100 μm. **e** Immunofluorescence staining for Calponin, GFP (indicating the mNCC) and DAPI. Scale bar: 100 μm. **f** Immunofluorescence staining for GFP (indicating the mNCC), S100 and DAPI. Scale bar: 100 μm. **g** Immunofluorescence staining for hNuclei, caspase3 and DAPI (top) and hNuclei, Ki67 and DAPI (with phase contrast) (bottom). Arrows indicate cells positive for caspase3 or Ki67. Scale bar: 100 μm. **h** Percentage of Ki67+ and caspase3+ cells per field identified in the muscle layer of multiple sections of recellularized scaffolds after omental transplantation. Data: mean ± SEM (n = 3). **i** Immunofluorescence staining for hNuclei, vWF and DAPI (with phase contrast). Sc: scaffold; om: omentum; dotted line indicates the separation border between omentum and scaffold. Scale bar: 100 μm. **j** Images of control (unseeded) scaffolds 7 days after implantation in the omentum. Hematoxylin and eosin staining (left – scale bar: 250 μm) and immunostaining for hNuclei, SM22 and DAPI (with phase contrast) (right – scale bar: 100 μm). Sc: scaffold; om: omentum; dotted line indicates the border between omentum and scaffold. **k–o** Representative images of scaffolds seeded with hMAB + mFB + mNCC, cultured in dynamic conditions for 11 days, implanted in the omentum of NSG mice, harvested after 7 days and re-populated with ROEC seeded on the luminal side. Scaffolds were analysed at 3 and 7 days of culture. Immunofluorescence for Ki67 and E-cadherin (**k**), CK14 and CK13 (**l**), CK14 and p63 (**m**), Pancytokeratin and hNuclei (**n**), and caspase3 (**o**). Nuclei were stained with DAPI. Insets show specificity and localization of positive cells. Scale bar: 100 μm (**j**, **l–n**) and 50 μm (**k**)

covered the entire surface (1–3 cm$^2$) of the scaffold. Importantly, following 2 weeks in culture and upon stratification, ROEC remained able to proliferate symmetrically and asymmetrically thus maintaining a functional basal layer, whilst expressing tight-junction protein ZO-1 in the uppermost layer. This indicated that ROEC might maintain self-renewal capacity despite functional differentiation and this capacity will be crucial for future clinical translation of the proposed approach.

The use of dynamic culture conditions (using a custom glass bioreactor) allowed easy localization of luciferase+ cells during cell seeding and tissue growth. Bioluminescence imaging was advantageous compared with post-fixation immunohistological analysis, producing live quantitative information on cell distribution and viability and could become an invaluable tool through all phases of dynamic culture up to in vivo transplantation.

In vivo implantation of a multi−layered oesophagus requires a vascular supply to allow cell survival and functional maturation. Herein, pre-vascularization was obtained by means of placing the construct in the omentum of mice for 1 week. The omentum is extremely angiogenic and has been used to improve tissue maturation by stimulating graft vascularization[7,37,57,58]. Survival of hMAB and mNCC within the construct after omental implantation indicates that in vivo conditions provided nutrients and oxygen to the graft. Notably, prompt vascularization was achieved after 1 week in vivo with minimal tissue remodelling so that ROEC were still capable of adhering and stratifying when seeded in the lumen of the tubular scaffolds post-transplantation. This approach could be adopted for clinical translation with heterotopic transplantation of a muscle conduit allowing vascularization prior to delivery of the epithelial layer, followed by orthotopic transplantation to substitute the oesophagus. In a similar study performed implanting porcine decellularized scaffolds in the omentum of rats, the in vivo maturation and pre-vascularization phase seemed to have a negative effect on elasticity of the decellularized scaffolds[38]. This difference suggests caution on using this approach and further tests of mechanical properties post-in vivo should be considered. An expected host inflammatory reaction was detected after omental implantation, with infiltration of macrophages and neutrophils in the *muscularis externa* of the scaffold. This result was in line with published studies that highlighted an initial inflammatory response to engineered grafts with decellularised scaffolds. The acute host response to ECM-derived scaffolds activates the innate immune system, including neutrophils and macrophages. These scaffolds have been associated with a robust but favourable host immune response that triggers constructive remodelling outcomes and an overall anti-inflammatory environment. In particular,

macrophages have been recognized as a critical determinant of regeneration after injury, tissue remodelling and cross-talk with endogenous stem/progenitor cells[59,60].

Importantly, we show here that the oesophageal muscle conduit could also be cryopreserved prior to transplantation, overcoming one of the key bottlenecks for preclinical and clinical translation of tissue engineering[61]. The possibility of storing a bio-engineered tissue at any stage of the bio-fabrication process could be a major advance to make the graft readily available for surgical implantation. This can be particularly relevant on a complex organ such as the oesophagus where multiple cell types are necessary. Moreover, it resolves the problem of transporting the engineered organs. At the moment transport is difficult with limitations related to distance (because of the time allowed outside the incubator), risks related to cell viability and contaminations. Having a cryopreserved organ increased the possibility of commercialization and benefit to a larger number of patients. Slow-cooling cryopreservation has already been used for successful preservation of decellularised scaffolds including the oesophagus[29,62,63], but has never been applied to cell-seeded scaffolds. The same protocol was used here to cryopreserve constructs, after the first stage of the bio-engineering process, for 2 weeks in liquid nitrogen. In this pilot experiment, cells within the construct recovered in culture after thawing, displaying further proliferative capacity; however, future investigations will be required to fully explore the potential of cryopreserved oesophageal constructs.

The multi-stage approach was planned taking into consideration from the beginning the potential of clinical translation. To this aim, considering the possible application to patients with oesophageal atresia, human cells were isolated from tissues, which will normally be approached during clinical standard procedures to those patients. Mesoangioblasts and fibroblasts have been isolated from skeletal muscle biopsies, which can be provided from the abdominal muscle wall of those patients at birth in case of a gastrostomy placement. These biopsies have been retrieved aseptically and subsequently processed such that they can be translated directly into a GMP-compliant manufacturing process. During the same procedure, neural crest cells (isolated from gut biopsies in this study) can be derived directly from the stomach. Finally, the oesophageal epithelium can be derived from an endoscopic biopsy of the oesophagus during the initial assessment of the oesophagus at birth. The challenge remains to complete the ex-vivo production of this oesophageal implant and to test it in a large animal model. The subsequent translation to first-in-man and then routine production will be greatly facilitated by the design and manufacture of a bespoke "closed-system" bioreactor for decellularization and then recellularization of donor

oesophageal scaffolds. Another feature of our work is the use of human and non-human cell populations. The use of different species facilitated both their identification in co-culture experiments and the analysis of their fate, and allowed using neuronal precursors from transgenic animals. It also allowed demonstration that each cell type, expanded in different culture conditions, maintained its lineage specification and integrated functionally in the combined structure. For a successful pre-clinical translation, all-human cells repopulated grafts will have to be developed and tested. Although we believe that cell-cell and cell-matrix interactions described in this study will promote comparable organ regeneration effects, testing all-human constructs will be essential to determine safety and functional outcomes for clinical translation. Decellularized oesophageal scaffolds alternative from rat origin will have different composition in muscle fibres since rodent oesophagi contain only skeletal muscle. The potential effect of this change in muscle layer composition in cell behaviour in the scaffold should also be considered in future investigations. Additionally, it is important to evaluate scalability of our approach and we are investigating the two-stage recellularization process in a large animal model for functional studies to replace a full-circumferential oesophageal defect. In vivo preclinical studies would also provide essential information regarding the host immune response to the graft and ECM remodelling. In vivo functional tests after orthotopic replacement will be important to identify the long-term contribution of seeded cells to oesophageal regeneration. In a recent study, unseeded decellularized porcine oesophagi were used to replace a full thickness circumferential defect of the abdominal esophagus in pigs[38]. Mixed clinical outcomes were registered five weeks post-implantation, with evidence of vascularization in the muscle layer and submucosa, strong remodelling of the external layer and inflammatory cell infiltration.

In summary, with the present study we reconstructed a layered full-circumferential oesophageal tissue in vitro and propose its future use as a potential alternative treatment for oesophageal atresia or other oesophageal defects, where tissue replacement is needed. The approach here described could be easily scaled-up to larger constructs in the view of future pre-clinical experiments in larger animal models where all components are human.

## Methods

**Animals**. All animal procedures were in accordance with ethical approval and UK Home Office Project Licence PPL 70/7622 and 70/7478. 200–300gr Sprague Dawley male rats were used for oesophageal scaffolds and epithelial cells isolation. Mouse fibroblasts were isolated from C57BL/6 J mice and neural crest cells were obtained from transgenic Wnt1-cre;R26R$^{YFP/YFP}$ mice.

**Decellularization of rat oesophagi**. With a midline incision the oesophagus was harvested from the cervical portion to the gastroesophageal junction, washed with Phosphate Buffered Saline containing 1% antibiotic/antimycotic (PBS/AA; Sigma), measured and canulated. Oesophagi were decellularized with two cycles of detergent-enzymatic treatment (DET) according to established protocols[26,28]. Briefly, the oesophageal lumen was perfused with continuous fluid delivery (iPumps) at 1 ml min$^{-1}$. Each DET cycle was composed of deionised water at 4 °C for 24 h, 4% sodium deoxycholate (SDC; Sigma) at room temperature (RT) for 4 h and 2000 Kunitz DNase-I (Sigma) in 1 M NaCl at RT for 3 h. The process was repeated for two cycles. Decellularized oesophagi were sterilized with gamma irradiations.

**DNA quantification**. PureLink Genomic DNA MiniKit (Invitrogen) was used to isolate DNA following the manufacturer's instructions as previously described[26,27]. DNA samples were measured spectrophotometrically (Nanodrop).

**Histology**. Samples were fixed in 4% Paraformaldehyde (PFA; Sigma), dehydrated in graded alcohols, paraffin embedded and sectioned at 5 μm. Tissue slides were stained with Haematoxylin and Eosin (H&E; Leica) and Masson's Trichrome (MT; Leica, Raymond A Lamb, BDH Chemicals Ltd).

**Immunofluorescence and immunohistochemistry**. Tissue samples were fixed in 4% PFA, washed in PBS, dehydrated in 30% sucrose overnight, embedded and frozen in O.C.T. (Sakura Finetek) with ice-cold isopentane (Sigma) and stored at −80 °C. 7–10 μm thick sections were cut (Leica cryostat, UK), and slides stored at −20 °C. Cells and sections were permeabilized with 0.5% Triton X-100 in PBS for 10 min at room temperature, washed and blocked with 5% Goat Serum for 1 h at room temperature. Primary and secondary antibodies were diluted in 1% Goat Serum/PBS/0.01% Triton X-100. mNCC staining was performed using 1% BSA/ 0.15% Glycine (Sigma). Sections from in vivo experiments were labelled using a kit specifically designed to reduce endogenous IgG staining (M.O.M, Vector). All primary antibodies were applied overnight at 4 °C. Slides were incubated with Alexa Fluor secondary antibodies (Invitrogen) for 45 min at room temperature, washed and mounted with Vectashield + DAPI (Vector Labs). Cells were incubated with Hoechst diluted 1:1000, washed and maintained in PBS until imaged. List of primary and secondary antibodies used is shown in Supplementary Table 1. Human oesophageal tissue was used for comparison and obtained from a pediatric patient, with informed consent, during surgery at the Great Ormond Street Hospital, London, in accordance with ethical approval by the NHS Research Ethics Committee, REC Ref: 04/Q0508/79. The Committee was constituted in accordance with the Governance Arrangements for Research Ethics Committees and complied fully with the Standard Operating Procedures for Research Ethics Committees in the UK.

Paraffin-embedded sections were pressure-cooked for 3 min in Sodium Citrate Buffer (pH 6.0) followed by incubation in peroxidase blocking solution for 5 min and washing in Tris Buffered Saline pH 7.6 for 5 min. Following 1 h incubation, primary antibodies (listed in Supplementary Table 1) were detected using an avidin-biotin-based system (Vector).

Images were acquired with a Zeiss LSM 710 confocal microscope (Zeiss) and processed using ImageJ and Adobe Photoshop. Manual cell counting was performed by 3 operators in blind. Some tissue sections were imaged using an In Cell (GE Healthcare) high-content scanner and analysed to produce heat-maps of seeding density. hMAB and mFB proportions were calculated as the number of hNuclei$^+$ and hNuclei$^-$ cells over the total number of DAPI$^+$ cells in random sections from different regions of recellularized scaffolds immunostained for hNuclei and DAPI.

**Quantification of elastin and glycosaminoglycans**. Elastin and glycosaminoglycan were quantified as previously described[26].

The elastin content of native and decellularized tissues was quantified using the FASTIN elastin assay (Biocolor) according to the manufacturer's instructions. Briefly, the samples were homogenized and elastin was solubilized in 0.25 M oxalic acid. Two consecutive incubations were performed at 95 °C to ensure complete extraction of elastin. Extracts were incubated with 5,10,15,20-tetraphenyl-21H,23H-porphine tetrasulfonate (TPPS) dye and absorbance was determined at 555 nm spectrophotometrically (Tecan Infinity). Elastin concentrations from a standard curve were used to calculate the elastin content of the tissue.

The Glycosaminoglycan (sGAG) content of native and decellularized tissues was quantified using the Blyscan GAG Assay Kit (Biocolor). Briefly, the tissues were digested with papain at 65 °C for 18 h and aliquots of each sample were mixed with 1,9-dimethyl-methylene blue dye and reagents from the GAG assay kit. The absorbance at 656 nm was measured spectrophotometrically (Tecan Infinity) and compared to standards made from bovine tracheal chondroitin-4-sulfate to determine the sGAG content.

**Biomechanical testing**. Biomechanical properties of native and decellularized oesophagi were evaluated using a uniaxial testing machine Synergie 200 H MTS (Synergie). Briefly, flat shaped specimens were preconditioned with 8 cycles of loading-unloading up to 70% strain at a constant rate of 0.1 mm min$^{-1}$, then 4 cycles of loading-unloading up to 70% strain at a constant rate of 0.1 mm min$^{-1}$ were performed to evaluate the stiffness of the samples. Afterwards, a relaxation test was performed: specimens were loaded at a constant rate of 0.3 mm min$^{-1}$ and left to relax at 40% strain for 100 s. A failure test was finally carried out with at a constant rate of 0.1 mm min$^{-1}$.

**SEM**. Unseeded and seeded samples were fixed in 2.5% Glutaraldehyde (Sigma) 0.1 M Phosphate Buffer for 24 h at 4 °C and SEM performed as previously described[26].

**Primary cell cultures**. Human mesoangioblasts (hMAB) were isolated from paediatric skeletal muscle biopsies from patients aged from 1 week to 8 years old, with informed consent, during surgeries at the Great Ormond Street Hospital, London, in accordance with ethical approval by the NHS Research Ethics Committee, REC Ref: 11/LO/1522. The Committee was constituted in accordance with the Governance Arrangements for Research Ethics Committees and complied fully with the Standard Operating Procedures for Research Ethics Committees in the UK. Cells were isolated according to a previously published protocol[45] with modifications.

Biopsies were rinsed in sterile PBS, dissected into small pieces (~ 2 mm), removing possible adipose tissue, and seeded on petri dishes coated with Matrigel (growth factors reduced, BD) diluted 1:100 to favour attachment and cell

outgrowth. Muscle fragments were covered with proliferation medium [Megacell medium (Sigma), 5% Fetal Bovine Serum (FBS, Gibco), 1% non-essential aminoacids (Gibco), 1% L-Glutamine (Gibco), 1% Penicillin-Streptomycin (Gibco), 0.1 mM ß-mercaptoethanol (Sigma) and 5 ng ml$^{-1}$ bFGF (Sigma)] and incubated at 37 °C, 5%O$_2$ and 5%CO$_2$. Outgrowths with high cell density were collected through trypsinization and transferred to flasks (Nunc). Muscle fragments were re-plated up to 4 times till complete depletion occurred. Mesoangioblasts were passaged at 60–70% confluence for up to 10 passages and analysed for expression of pericyte-like markers and differentiation potential. For smooth muscle differentiation, hMAB were incubated in High Glucose DMEM (Gibco) supplemented with 1% L-Glutamine, 1% Pen-Strep, 2% Horse Serum (Gibco) and 5 ng ml$^{-1}$ TGFβ1 (Sigma) for 7 or 14 days, with fresh TGFβ1 provided daily.

Mouse fibroblasts (mFB) were isolated from hindlimb skeletal muscles (extensor digitorium longus) by enzymatic digestion with 0.1% Collagenase type I (Sigma) for 80 min at 37 °C, plated onto 100 mm dishes pre-coated with 2% Horse Serum and incubated at 37 °C, 20%O$_2$ and 5%CO$_2$. After removal of floating cells and fragments, cells were maintained in culture until confluent using growth medium consisting of High Glucose DMEM, 20% FBS, 1% L-Glutamine and 1% Pen-Strep.

Mouse neural crest cells (mNCC) were isolated from a reporter mouse line, Wnt1-cre;R26R$^{YFP/YFP}$, as previously described[54]. Briefly, YFP$^+$mNCC (enteric nervous system stem cells, neurons and glial cells) were obtained from postnatal (P2-P7) mice following enzymatic dissociation and FACS for YFP was performed using a MoFloXDP cell sorter (Beckman Coulter). The YFP$^+$ cells were plated onto fibronectin-coated 6-well dishes and cultured in DMEM F12 (Gibco) supplemented with B27, N2 (both Life Technologies), 20 ng ml$^{-1}$ EGF (Peprotech), 20 ng ml$^{-1}$ FGF (Peprotech) and Primocin antibiotic (In vivo Gen).

Rat oesophageal epithelial cells (ROEC) were isolated from the oesophagus of Sprague Dawley male rats. The muscle layer was carefully removed and the submucosa and mucosa layers were enzymatically dissociated using 0.25% Trypsin/EDTA (Sigma) at 37 °C, until a single cell suspension was obtained. Dissociated cells were plated on a layer of lethally irradiated mouse 3T3-J2 cells and cultivated as previously described[64,65].

Human epidermal growth factor (hEGF, 10 ng ml$^{-1}$, PeproTech) was added to medium after 4 days and then at each feeding. Cells were passaged every 6–7 days. ROEC were successfully isolated from 6 rats and expanded for at least 10 passages. Colony forming efficiency (CFE) assay was performed by plating 100 or 300 cells in a 60 mm culture dish with irradiated feeder layers.

**Alkaline phosphatase staining.** Alkaline phosphatase staining was performed according to the manufacturer's protocol (Sigma). Cells in 35 mm dishes were washed, fixed for 30 sec using a Citrate/Acetone/Formaldehyde solution, washed for 45 sec with deionized water and then incubated with the working solution for 20 min in the dark, at room temperature.

**FACS.** Cells were detached with a cell scraper and washed in sterile Hank's Balanced Salt Solution (Sigma) supplemented with 2% FBS. $5 \times 10^4$ to $1 \times 10^5$ cells were used for each staining.

Surface proteins were labelled with primary antibodies as listed in Supplementary Table 2 for 20 min at 4 °C, protecting samples from direct light. After washing with HBSS/2% FBS, cells were resuspended in the same buffer for analysis using BD LSRII Flow Cytometer. Data were analysed collecting $5–10 \times 10^3$ events for each sample and using the software FlowJo. As negative control, unstained cells were used. As positive control, compensation beads (Thermo Fisher) with antibodies or single-stained cells were used.

**Seeding of cells.** hMAB, mFB and mNCC were trypsinised and resuspended in a solution of PBS, 0.5 ng ml$^{-1}$ Collagen type I (Sigma) and 0.1 ng ml$^{-1}$ Fibronectin (Sigma) and kept on ice till seeding. The conditions were: hMAB alone, hMAB + mFB (ratio 85:15) and hMAB + mFB + mNCC (ratio 57:10:33). The volume of cell suspension was calculated to inject $1 \times 10^6$ cells every 5 mm length of scaffold for hMAB alone and hMAB + mFB conditions, or $1.5 \times 10^6$ cells for hMAB + mFB + mNCC condition. 3–3.5 cm of rat oesophageal decellularized tubular scaffolds was seeded. Cells were either dropped in 10 μl droplets on the external surface of decellularized scaffolds or microinjected every 3–4 mm with an insulin syringe (MyJector)/27 G needle at multiple sites along 3 distinct longitudinal lines. Multiple microinjections were performed manually under a stereomicroscope to ensure cell delivery to the muscle layer. For easy handling and ensuring constant tight tension whilst microinjecting, a 6 F nasogastric tube (Enteral) was inserted into the scaffold first. The intrusion of the plastic tube also removed the mucosal layer from the luminal side of the scaffold, as required for future seeding of epithelial cells. Multiple injection seeding was used in all the experiments showed in the main figures, after initial comparison with surface seeding.

Cultivated ROEC were seeded on the luminal surface of the oesophageal scaffold. Cell density for ROEC seeding was $1.3 \times 10^5–2.6 \times 10^5$ cells cm$^{-2}$, depending on the culture conditions.

**Static and dynamic culture.** For static cultures, seeded scaffolds were placed in multiwell plates and cultured for 6 ÷ 9 to study cell distribution and migration, and

11 days to assess cell proliferation and differentiation. For dynamic cultures, seeded scaffolds were sutured to glass rods with 3–0 silk sutures and these were placed in a custom dual glass chamber. This chamber allows physical separation between lumen and external surface of the scaffold (as shown in Fig. 2b). A schematic representation of the bioreactor and its components used for dynamic culture is reported in Fig. 2a. The bioreactor was designed to allow medium flow inside the lumen whereas an inlet and outlet present in the external chamber allowed medium flow around the scaffold. Lumen and external flows were controlled with an Applikon® bioreactor, connected to a reservoir of medium. The external chamber was filled up with medium and connected to the Applikon bioreactor. Medium flow was activated 6 h after seeding. Medium flow was 5 ml min$^{-1}$. Glass chambers and medium reservoirs were maintained in a humidified incubator at 37 °C and 5% CO$_2$. Medium was changed entirely every 2–3 days. Scaffolds were cultured for 2 days in proliferation medium (PM) and 9 days in smooth muscle differentiation medium, with TGFβ1 added daily through the injection point of the bioreactor.

For epithelial cell seeding in static conditions, scaffolds were slit open and the mucosa layer was removed. Flat oesophageal scaffolds were cut into pieces (~0.5–1.0 cm$^2$) and placed over a transparent cell culture insert (PET membrane, 0.4μm pore, Greiner Bio-One) in a 6-well culture plate. After cell seeding, the scaffolds were incubated at 37 °C for approximately 1 h. Then cFAD medium was added in the plate, enough to touch the membrane, and on top to cover scaffolds. Fresh medium was added daily to ensure that the scaffolds were continuously covered. After 7 days, the level of the medium was reduced to promote stratification of seeded cells. Seeded scaffolds were kept in culture up to 2 weeks. For seeding in the bioreactor, cells were delivered in the lumen of the oesophageal scaffold with a thin cannula through the rods of the bioreactor. The chamber was incubated at 37 °C for 2 h without flow and rotated every 30 min. Medium flow started after 24 h from seeding, at a speed of 0.5 ml min$^{-1}$.

**Polar diagram analysis.** Whole histological sections were imaged using an InCell Analyzer 2200 (GE Healthcare Life Sciences) at 20x magnification and excitation wavelengths corresponding to FITC, TexasRed and DAPI. Fluorescence normal-isation was applied following background subtraction to ensure a homogeneous fluorescence level across the whole tissue section. Global thresholding was used to generate a tissue mask. An in-house image processing automated the detection of nuclei, which was validated against manual detection. Fluorescence intensity was integrated in a defined region around the position of each nucleus, across all relevant channels. Image cytometry techniques were used to manually separate positive from negative cells. The distribution of the resulting cell populations within the tissue was analysed by calculating the local cell density. From this processing, a set of graphical and quantitative outputs were generated. Cell distribution maps show the physical location of cells within a tissue section. Polar distributions represent the location of the cells in a standardised format, which were used to compute cell density maps showing cell density within the scaffold and in relation to the lumen and the external surface. The percentage of cells positive for the various markers was also calculated.

**Migration assay.** The Vybrant MTT colorimetric assay (Life Technologies, Thermo Fisher) was used to determine viable cell migration. Tubular scaffolds seeded with either $8.5 \times 10^5$ hMAB or $1 \times 10^6$ hMAB with/without mFB (ratio 85:15, $n \geq 3$) were cultured in static conditions for 6 days and then incubated with MTT diluted 1:10 in proliferative medium for 4 h at 37 °C. Scaffolds were washed with PBS, opened longitudinally and placed at the bottom of a multiwell plate for imaging with a stereomicroscope. Formazan-positive cells were visible within the scaffolds. Images of flat open scaffolds were analysed for cell migration measuring the colour intensity along 8 random lines drawn radially from the centre of cell clusters to the edge of the scaffold using ImageJ software. The Gray Value graph obtained from the lines was used to measure the distance in mm covered by the cells, calculated between the centre of the cluster and the most distant cell. Analyses and calculations were performed by three independent operators in blind on 6 scaffolds per condition and in 2 experimental replicates. The average cell migration distance in seeded scaffolds was determined.

**Lentivirus preparation for bioluminescence imaging.** The lentiviral vector pHIV-LUC-ZsGreen (Addgene Inc. MA, USA, Plasmid #39196, kind gift from Dr. Bryan Welm, Department of Surgery, University of Utah) was used to generate a lentivirus containing both ZsGreen fluorescent protein and firefly luciferase from an EF1-alpha promoter. Briefly, LUC-ZsGreen lentivirus was produced by co-transfecting 293 T cells with the above plasmid along with packaging vectors [pRSV-Rev (Addgene Plasmid #12253), pMDLg/pRRE (Addgene #12251) and VSV-G envelope plasmid pMD2.G (Addgene Plasmid #12259)]. Transfection was performed according to manufacturer's instructions for 6 h at 37 °C. The medium (DMEM containing 10% FBS, Gibco) was exchanged for virus collection. After 24 h, the virus-containing medium was purified by centrifugation at 2500 rpm (4 °C) and filtered through a 0.45 μm membrane. Medium was ultracentrifuged at 50.000 g for 2 h at 4 °C (SW28 rotor, Optima LE80K Ultracentrifuge, Beckham). The viral pellet was resuspended in 100 μl pre-cooled serum-free DMEM (Gibco), aliquoted and stored at −80 °C. Viral titres were calculated by transducing HeLa cells. Transduction efficacy was determined by flow cytometric analysis and measured as

the proportion of cells expressing the fluorescent protein ZsGreen 72 h after transduction.

hMAB and ROEC were transduced with the lentivirus as described above and FACS sorted (FACS Aria, BD Bioscience) to get a pure population. Sorted cells were expanded and used for downstream experiments.

**Bioluminescent imaging**. Bioluminescence was detected using an IVIS Lumina Series III Pre-clinical In Vivo Imaging System (IVIS; Caliper Life Sciences) and Living Image 3.2 software (Caliper Life Sciences). All images were taken on either stage C or D, with automated aperture setting, 1 min exposure time and using a small binning (resolution). Twenty minutes prior to imaging, culture medium was exchanged with medium containing 150 µg ml$^{-1}$ D-Luciferin for culture plates or 150 µg ml$^{-1}$ D-Luciferin was injected into the internal and/or external chamber of the bioreactor via 3-way luer taps and imaged as described above. Stage D was used for zoomed out images of the entire reactor and stage C for all other images and analysis.

**Bioluminescent image analysis**. All images were analysed using Living Image 3.2 software (Caliper Life Sciences). This generated pseudo-coloured scaled images overlaid on grey scaled images, providing 2-dimensional localization of the source of light emission. Regions of interest (ROI) were selected using shape drawing tools and the light emission was quantified in photons s$^{-1}$. ROI shapes were kept constant between images within each experiment.

**Cryopreservation of seeded scaffolds**. Scaffolds seeded with Luc$^+$ZsGreen$^+$hMAB and mFB were cryopreserved after 8 days of static culture following an established protocol[29]. Briefly, samples were cooled slowly ($-1$ °C min$^{-1}$) while immersed in 40% PM, 50% FBS and 10% Dimethyl Sulfoxide (Sigma). Slow cooling was achieved at $-80$ °C overnight. Samples were stored in the vapour phase of liquid nitrogen for 2 weeks. They were thawed rapidly in a 37 °C water bath and incubated in PM for 24 h before being cultured for additional 7 days in static conditions using DM. Scaffolds were imaged and analysed with IVIS at specific time points.

**Oxidative metabolism**. $CO_2$ production was used as a marker of mitochondrial oxidative metabolism[53]. Following 3D-cultures, 0.5cm-long repopulated scaffolds were placed inside 12-well dishes and covered with 1 ml of glucose-free medium supplemented with 17.5 mM [U-$^{13}$C] Glucose (Cambridge Isotope Laboratories). For 2D cultures, hMAB were plated with/without mFB onto 35 mm dishes and differentiated towards smooth muscle cells before receiving 3 ml of the same medium.

To prevent $CO_2$ evaporation, mineral Oil was added before incubating the samples for 6 h at 37 °C; 100 µl of medium was removed from each sample every 2 h and stored in a tightly sealed glass tube (Examiner, Labco) at $-20$ °C until analysis. For analysis, samples were thawed at room temperature and 100 µl of 1 M HCl was injected through the septa of the glass tubes to release $CO_2$ from bicarbonate. After shaking, vials were loaded on GC-IRMS racks and analysed by isotope mass spectrometry. The ratio of $^{13}CO_2$ to $^{12}CO_2$ produced is proportional to the amount of oxidized glucose[53].

**Calcium imaging**. Recellularised oesophagi were immersed in previously oxygenated (95% oxygen/5% carbon dioxide) Krebs solution (in mM: 120.9 NaCl, 5.9 KCl, 1.2 MgCl2, 2.5 CaCl2, 11.5 glucose, 14.4 NaHCO3 and 1.2 NaH2PO4). Tubular oesophagi scaffold preparations were pinned tightly, serosal side up, in a Sylgard-lined chamber. Tissues were then loaded with the fluorescent Ca2 + indicator Rhod-3 (ThermoFisher Scientific; 5 mM) and Cremophor EL (Fluka Chemika, Buchs; 0.00001%) in Krebs solution at room temperature for 20 min with continuous oxygenation. Tissues were washed ($2 \times 10$ min, Krebs) prior to imaging. Subsequently, transplanted YFP$^+$mNCC-derived cells were identified and live fluorescence imaging was performed on an Olympus BX51 microscope equipped with 20x water dipping lens (XLUMPlanFL N, NA 1, Olympus Europa) and an EMCCD camera (iXon Ultra 897, Andor Technology,). Rhod-3 was excited at 530 nm using an OptoLED (Cairn Research Limited) and fluorescence emission was collected at 605/70 nm. Images ($512 \times 512$ pixels2) were acquired at 2 Hz. Electrical train stimulation (2 s, 20 Hz of 300 ms electrical pulses; Electronic stimulator 1001, AD instruments) was applied via a platinum/iridium electrode (tip diameter 2–4um, World Precision Instruments), placed at a distance of 200µm from the observed ROI. Electrical point stimulation was applied as described above and images were collected using OptoFluor software (Cairn Research Limited). Post-acquisition analysis was performed in Fiji[66]. Movement artifacts were removed by registering the image stack to the first image. ROI were drawn over each cell, fluorescence intensity was normalised to basal fluorescence for each ROI (F∆/F0), and peaks analysed. Images were pseudo-coloured using Fiji Lookup tables (Hot red) for presentation as representative figures.

**In vivo transplantation models**. Live animal work was ethically approved and carried out under Home Office Project Licence PPLs 70/7622 and 70/7478. NOD-

SCID-gamma (NSG) mice were anaesthetized with a 2–5% isoflorane:oxygen gas mix for induction and maintenance.

For transplantation in omentum, Buprenorphine 0.1 mg Kg$^{-1}$ was administered at the induction for analgesia. Under aseptic conditions a midline laparotomy was performed. The stomach was externalised from the incision and the omentum stretched from the great curvature. A segment of the engineered scaffold ($n = 3$) or unseeded scaffold ($n = 2$) was then enveloped in the omentum, using a 8/0 prolene suture to secure the closure of the omental wrap. The stomach and the omentum were replaced in the abdomen and the laparotomy closed using 6/0 Vicryil. Animals were allowed to normally eat and drink immediately after surgery and no further medications were administered during the post-operative period. At 7 days post-transplantation, animals were euthanized and the scaffolds harvested together with the omental envelope. The non-absorbable prolene stitches were used to identify samples.

For kidney capsule transplantation, 5 mm seeded ($n = 4$) or unseeded ($n = 2$) oesophageal scaffolds were transplanted under the kidney capsule of adult NSG mice[67] which were sacrificed at 7 days for analysis.

**Statistics**. Unless stated otherwise, the n value reported in the manuscript for each analysis/assay represents the number of biological replicates. Data are expressed as mean ± SEM unless otherwise stated. Significance was determined by one-way ANOVA and Tukey's multiple comparison test or Kruskal-Wallis and Dunn's multiple comparison test; ANOVA with post-hoc Bonferroni test and two-tailed unpaired Student's $t$-test. A $p$-value of less than 0.05 was considered statistically significant. Statistical analysis was performed using GraphPad Prism 6 (GraphPad Software).

## Data availability
Data are available from the corresponding authors upon reasonable request.

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

## Acknowledgements

P.D.C. is supported by National Institute for Health Research (NIHR-RP-2014-04-046). L.U. was supported by NIHR and the OAK Foundation (W1095/OCAY-14-191). S.E. is supported by Great Ormond Street Hospital (GOSH) Children's Charity. P.B. is supported by the UCL Excellence Fellowship Programme, the Rosetrees Trust (M362, M553) and the NIHR Biomedical Research Centre at Great Ormond Street Hospital for Children NHS Foundation Trust. A.U. was supported by the OAK Foundation (W1095/OCAY-14-191). G.C. is supported by the MRC (MR/P016006/1). D.M. is supported by the NIHR GOSH BRC award 17DD08. C. Camilli was supported by UCL Grand Challenge Studentship. This study was supported by the UK Stem Cell Foundation, the Cell and Gene Therapy Catapult, the GOSH Charity (V1282) and the OAK Foundation. NIHR support was dedicated exclusively to the human cells used in the paper. All research at Great Ormond Street Hospital NHS Foundation Trust and UCL Great Ormond Street Institute of Child Health is made possible by the NIHR Great Ormond Street Hospital Biomedical Research Centre. The views expressed are those of the author(s) and not necessarily those of the NHS, the NIHR or the Department of Health. The authors would like to thank Silvia Perin, Ayad Eddaoudi, Simone Russo and Colin Butler for the technical help and support. We also thank Venizelos Papayannopoulos (Francis

Crick Institute) for the gift of antibodies. P.D.C. thanks TOFS and EATS Charities who support babies born unable to swallow.

## Author contributions

L.U., A.U., N.T., M.L., G.C., P.B., P.D.C designed the work. L.U., C.Camilli., D.E.P., C. Crowley, D.N., F.S., P.M., C.J.M.C., A.F.P., S.F.A., M.C.S., D.K., E.H., K.D., M.T., R.R.W., M.O.B., D.M., S.K., A.V., A.G., S.E., P.B. performed the experiments and contributed to data collection. L.U., C.Camilli, D.E.P., C.Crowley, D.N., C.J.M.C., A.F.P., S.F.A., M.C.S., M.O.B., D.M., S.L., S.E., N.T., M.L., G.C., P.B. and P.D.C. contributed to data analysis and interpretation. L.U. prepared the figures. L.U., C.Camilli, D.E.P., D.N., C.J.M.C., P.B. and P.D.C. wrote the manuscript. C.Crowley, F.S., A.U., S.M., S.E., N.S., N.T., M.L., G.C., P.B. and P.D.C. contributed to the revision of the manuscript.

## Additional information

**Competing interests:** P.D.C., L.U., and A.U. are named inventors of patent application No. PCT/EP2016/071114 and P.D.C., L.U., and C. Crowley are named inventors of UK patent application No. 1708729.7. The remaining authors declare no competing interests.

