## [Peer Review File · Nature Communications]

Reviewers' comments:

Reviewer #1, expert in bioreactors and 3D cell culture scaffolds (Remarks to the Author):

In this study Urbani et al. propose a method to engineer oesophagus grafts by combining a rat decellularized oesophageal scaffold with human mesoangioblasts (hMAB), mouse fibroblasts (mFB), murine neural crest cells (mNCC) and rat oesophageal epithelial cells (ROEC) through sequential in vitro and in vivo phases. A bioreactor is used to improve cell distribution and enhance cell differentiation.

The manuscript is overall well-written and the authors provide a proof-of-principle for the successful generation of tissue-engineered oesophagus. Each component of the study (i.e., oesophagus decellularization, re-population with multiple cell types, use of a bioreactor system, comparison with native tissue) represents in itself a rather incremental step forward in the field, of descriptive nature. What would be required for reaching a convincing level of relevance is the demonstration that combination of all such elements leads to a functional superiority of the resulting graft over existing strategies. The two-stage recellularization process is original and innovative, but again no proof of the need for such procedure to improve therapeutic outcome is offered. Thus, at least some of the perspectives listed at the end of the discussion (e.g., orthotopic implantation in a large animal model, host immune response to the graft, ECM remodelling, long-term contribution of seeded cells to oesophageal regeneration) should be included in a revised manuscript.

Additional major issues:

1. A certain number of controls is missing, in particular regarding stainings for differentiation assessment which should be systematically performed on both undifferentiated and differentiated cells (e.g. Figure 1c/d, Figure 5d/e/f)
2. The authors claim the use of a novel "customized" bioreactor. No details on the bioreactor structure/design/functionality/operational parameters are provided
3. If cellular colonization of scaffolds is a major hurdle, why was seeding not performed dynamically (under flow) using the bioreactor system?
4. The impact of cryo-preservation is not correctly assessed, as no quantification of cell death is performed (e.g. cas9, PI). Moreover, this was performed on a statically cultured scaffold -with a surprisingly limited cell distribution- while the dynamic condition is finally adopted. No evidences of a successful cryo-preservation of the final construct (after in vivo phase and ROEC seeding) are provided
5. Quantitative data should be systematically compiled when comparing static vs dynamic culture (e.g. Figure 4), in order to correctly assess spreading/engraftment/differentiation of mNCC in given conditions
6. The method is validated using 3 different cellular species (rat, mouse and human). The rationale behind should be clearly stated, and not solely justified by the ease of cellular identification within the grafts. This affects both clinical translation of the method, but also relevance of the presented data.
7. Neither the functionality nor the immunogenicity of the grafts is demonstrated.

Specific comments:

- line 78: the statement "decellularization preserves architecture and composition" should be tempered, as no protein quantification is included. It could be valuable to quantitatively assess if the decellularization method preserves to some extent ECM proteins, in particular those capable to play a role in the subsequent guidance of differentiation

- Figure 1e: displaying triplicates is confusing. The authors should pick 1 representative scaffold per condition

[ED: please move other examples to the supplement]

- Figure 2a: why not seeding also cells dynamically? If cells always seeded statically, dynamic culture does not increase engraftment, but rather proliferation of pre-seeded cells.

- Figure 3: it is not clear why some scaffolds were analysed after 6 days (figure 2) and others after 11 days. Does this correspond to 2 different phases of culture (e.g. 6 days of proliferation then differentiation till 11days)? Why not performing all analysis at the 11 days timepoint?

- Figure 3b: a comparative staining of native oesophageal muscle would be helpful.

- Figure 5j: it is not clear if cells were dynamically seeded, or seeded and then cultured in bioreactor.

- Figure 6: there is no comparison (even semi-quantitative) of epithelium maturation between the in vivo extracted and the in vitro decellularized scaffold from figure 5.

Ivan Martin and Paul Bourguine

Reviewer #2, expert in tissue engineering inc. oesophagus (Remarks to the Author):

Currently, there are two approaches to esophageal tissue engineering. One, the one undertaken by the authors, consists in constructing in vitro a substitute whose characteristics are as close as possible to the organ to be replaced. The other involves bringing into contact and implanting in vivo the elements considered necessary (extracellular matrix seeded with stem cells for example) so that tissue remodeling towards an esophageal phenotype occurs in vivo.

In the present work, 4 cell types are used for the making of the substitute, including 3 in co-culture (mesoangioblasts, fibroblasts and enteric neural cells). This work demonstrates that the co-culture of the first two cell types improves the distribution, homogeneity and migration of mesoangioblasts in the matrix; that cell colonization in the matrix is improved by dynamic incubation vs static incubation; that dynamic culture improves the differentiation of mesoangioblasts into smooth muscle cells; that it is possible to co-culture different cell types from different species; and finally that the substitute obtained by this approach has a structure close to that of the native esophagus, in vitro and after implantation in vivo. The approach is original because such a combination of cell types has never been tested in this context.

This work effectively analyzes the optimal conditions of construction of a finalized hybrid substitute. The amount of work is remarkable and the scientific demonstration rigorous, making this work convincing. No further evidence should be required to strengthen the conclusions of this paper. The level of detail from the experiments provided is adequate.

However, one can question the possibility of application in human clinic of such an approach, given its complexity (4 cell types, sequential cell seeding). The question to which the authors have the project to answer and which underlies the merits of this approach is quid of the cellular arrangement and the survival of these cell populations after circumferential replacement of the esophagus by this substitute.

Questions:

- is it not deleterious to differentiate mesoangioblasts before their implantation in vivo ? Is it not indeed preferable to preserve their ability to differentiate into other cell types and keep their capacity of secretions of factors involved in tissue regeneration, before implantation of the substitute in vivo ?
- what quality controls, apart from the structural analyzes, were carried out on the decellularized matrix (toxicity, immunogenicity, preservation of proteoglycans?).
- the future perspectives in terms of tissue engineering, as the "intelligent matrix", that contain factors implicated in tissue regeneration, without cell seeding, or other approach using cellularized matrix (see above). should be discuss
- what is the interest of cryopreservation of the substitute after cell implantation, when it is supposed that cellularized substitutes for clinical application will certainly use autologous cells.
- what is the reason for implantation of the substitute under the renal capsule. Are there differences in the maturation of the substitute between the two sites (large omentum and renal capsule)?
- Is there an inflammatory reaction at the implantation site?

In conclusion, favorable opinion for publication subject to modification of the discussion according to the remarks above

Reviewer #3, expert in oesophagus bioengineering (Remarks to the Author):

In this manuscript by Urbani and Camilli et al, the authors used a full thickness esophagus decellularized using a perfusion method of detergent-enzymatic treatment (DET), and repopulated with smooth muscle progenitor cells (hMAB), mouse fibroblasts (mFB), and murine enteric neural cells (mNCC); and allowed to differentiate in a static well setting or in dynamic setting with flow (bioreactor culture) for 11d. The triple cell-seeded scaffolds were then placed in the mouse omentum for 1 week to induce endogenous angiogenesis; and finally explanted and seeded with rat epithelial cells for 2 weeks. The authors also showed the ability to cryopreserve a cell-seeded scaffold with maintenance of viability of the seeded cells. Such a technique would be of interest to the regenerative medicine field and eventual clinical translation. The authors were relatively rigorous in their characterization of the scaffold; and of their cell types with immunolabeling, flow cytometry, and functional testing to determine the cell phenotype pre- and post-implantation as well as live cell imaging to determine the infiltration of the cells in the scaffold. However, the methods for scaffold decellularization were not provided and the metrics used to evaluate the extent of decellularization were inconsistent with established guidelines. The authors showed mature differentiation into skeletal muscle that was innervated, did show evidence of angiogenesis, and a mature epithelium, hence showing proof of concept of a tissue engineered, multi-layered esophagus.

Major revisions in this manuscript are required before it would be acceptable for publication.

Major comments:

1. The authors present an interesting approach. However, the barriers to clinical translation of this approach are significant and should be discussed. For example, this study utilizes 4 separate cell types that would need to be cultured for approximately five weeks prior to use. The cost of c-GMP facilities to ensure authenticity and sterility of each of the cultures; and especially the step implanting and explanting from the omentum, finally re-seeding after a surgical procedure. It would be too labor-intensive and cost-prohibitive for clinical translation. The cGMP considerations should be addressed as a limitation of the proposed approach. Do you have reason to believe this could be translated? The clinical translation would require the engineered esophageal construct to

be anastomosed to the native esophagus. Suture retention strength is not addressed and viability in vivo has yet to be established.

2. Figure 4d,e ◊ Where were different mNCC markers used in the dynamic versus static cultures? Were the same markers tried for both conditions?

3. Change "immunogenicity" to "antigenicity" line 284 pg 9 – ECM bioscaffolds will induce an inflammatory response, but need to acknowledge the difference between M1-like "pro-inflammation" and M2-like "constructive remodeling, immunomodulatory" activation of the response. ECM bioscaffolds, when appropriately decellularized, have been shown to induce the cellular response, but acellular ECM scaffolds have been shown to produce a more regulatory, anti-inflammatory phenotype as opposed to a pro-inflammatory phenotype when cells are part of the construct. (See Brown et. al.)

4. Many representative figures were included – at times it was difficult to follow because the static and dynamic figures were juxtaposed in different ways: sometimes static versus dynamic conditions were left to right, top to bottom or at angles to each other - but if there is a way to make it consistent from figure to figure, it would be easier to follow.

5. Controls showed inconsistencies. Comparisons were made between static versus dynamic for cell distribution, proliferation, skeletal muscle distribution, etc but implantation into the omentum was dynamic versus unseeded control, when that control was not shown previously; and then epithelial cells were finally seeded on both dynamic and static constructs. Why did the authors choose dynamic versus unseeded control instead of static for the omental implantation?

6. How were the n values derived for the different characterization experiments, was a power analysis performed? Please be consistent when stating biological and technical replicates for the cell-based assays, and fields of view that were quantified for immunolabeling for each n.

Minor comments:

7. The authors' review of the literature is deficient. Significant studies by Nieponice have shown promising results in preclinical animal studies and in human clinical studies. In addition, the authors failed to note the cohort study in which acellular scaffold materials successfully replaced neoplastic esophageal mucosa in five human patients. Although some of these studies were not full thickness defect, the authors should acknowledge these studies so that a comparison of cell-seeded versus acellular scaffolds for given indications can be evaluated.

8. How do you know the hMAB smooth muscle differentiation was triggered by TGFbeta (as stated in the text line 186 pg 6) when it was not a treatment in supplemental figure 4?

9. It should be stated in the results section that there were mature neurons and glial cells that were making connections – this shouldn't be brought up for the first time in the discussion line 336 pg 10

10. Pg 3 line 75 – change "but" to "and"

11. Pg 3 line 80 – "and successful" The introduction could use a revision, some awkward phrasing

12. Recommend changing "fresh" to "native" throughout manuscript to be more clear

13. Fig 3C (what is different between the left and right? Could use labels)

14. Labels wrong for fig 3f, g, h. line 184 pg 6?

15. Awkward sentence line 184 pg 6 "revealing...which revealed"

16. Line 184 first sentence of paragraph should be re-written, had to re-read several times to understand.

17. Figure 4B – only shows static (although refers to dynamic too in line 195, pg 6)

18. Figure 4f was confusing for me to tell which was what, could use labels

19. Not a sentence line 201 pg 6 "in addition..."

20. Spell out NSG line 241 pg 7

Citation for Comment 3: Brown, B. N., Valentin, J. E., Stewart-Akers, A. M., McCabe, G. P. & Badylak, S. F. Macrophage phenotype and remodeling outcomes in response to biologic scaffolds with and without a cellular component. *Biomaterials* 30, 1482–1491 (2009).

Reviewer #1, expert in bioreactors and 3D cell culture scaffolds (Remarks to the Author):

In this study Urbani et al. propose a method to engineer oesophagus grafts by combining a rat decellularized oesophageal scaffold with human mesoangioblasts (hMAB), mouse fibroblasts (mFB), murine neural crest cells (mNCC) and rat oesophageal epithelial cells (ROEC) through sequential in vitro and in vivo phases. A bioreactor is used to improve cell distribution and enhance cell differentiation.

The manuscript is overall well-written and the authors provide a proof-of-principle for the successful generation of tissue-engineered oesophagus. Each component of the study (i.e., oesophagus decellularization, re-population with multiple cell types, use of a bioreactor system, comparison with native tissue) represents in itself a rather incremental step forward in the field, of descriptive nature. What would be required for reaching a convincing level of relevance is the demonstration that combination of all such elements leads to a functional superiority of the resulting graft over existing strategies. The twostage recellularization process is original and innovative, but again no proof of the need for such procedure to improve therapeutic outcome is offered. Thus, at least some of the perspectives listed at the end of the discussion (e.g., orthotopic implantation in a large animal model, host immune response to the graft, ECM remodelling, long-term contribution of seeded cells to oesophageal regeneration) should be included in a revised manuscript.

We thank the Reviewer for her/his interest in our work and for the comments. Some of the topics listed have been added to the Discussion and explored in the comments below (point 6 and 7).

Additional major issues:

1. A certain number of controls is missing, in particular regarding stainings for differentiation assessment which should be systematically performed on both undifferentiated and differentiated cells (e.g. Figure 1c/d, Figure 5d/e/f)

We thank the Reviewer for the comment. Characterization of undifferentiated hMAB is shown in Figure 1c with immunofluorescence for specific mesoangioblast markers. Differentiation of hMAB towards smooth muscle after incubation with TGF β 1 is shown in Supplementary Figure 2b. A panel of immunofluorescence for smooth muscle markers on undifferentiated hMAB (without TGF β 1) has been added to Supplementary Figure 2b as control. The capacity of hMAB to fully differentiate into smooth muscle cells is shown by positivity for smoothelin with immunofluorescence after 2 weeks of exposure to TGF β 1. A representative image has been added to Supplementary Figure 2b.

Immunofluorescence of fibroblasts, reported in Figure 1d, identified cells as positive for Vimentin and Tcf4. Mouse NCC is a mixed population of undifferentiated cells, glia and neurons, as previously shown by the authors [55]. The expression of SOX10, S100 and TuJ1 by mNCC in culture was reported to confirm their phenotype before seeding experiments (Figure 4a).

Immunofluorescence of undifferentiated ROEC has been added in Supplementary Figure 6 to show absence of CK13 at day 4 of culture.

2. The authors claim the use of a novel “customized” bioreactor. No details on the bioreactor structure/design/functionality/operational parameters are provided

Thank you for the useful suggestion. We have added a simplified schematic of the bioreactor as Figure 2a to show the different components of the system. A detailed explanation of design and operational parameters has been added to the Methods (lines 630-656):

For dynamic cultures, seeded scaffolds were sutured to glass rods with 3-0 silk sutures and these were placed in a custom dual glass chamber. This chamber allows physical separation between lumen and external surface of the scaffold (as shown in Fig. 2b). A schematic representation of the bioreactor and its components used for dynamic culture is now reported in Fig. 2a. The bioreactor was designed to allow medium flow inside the lumen whereas an inlet and outlet present in the external chamber allowed medium flow around the scaffold. Lumen and external flows were controlled with an Applikon® bioreactor, connected to a reservoir of medium. The external chamber was filled up with medium and connected to the Applikon bioreactor. Medium flow was activated 6h after seeding. Medium flow was 5 ml/min. Glass chambers and medium reservoirs were cultured in a humidified incubator at 37°C and 5% CO₂. Medium was changed entirely every 2-3 days. Scaffolds were cultured for 2 days in proliferation medium (PM) and 9 days in smooth muscle differentiation medium, with TGFβ1 added daily through the injection point of the bioreactor.

[...] For seeding in the bioreactor, cells were delivered within the lumen of the oesophageal scaffold with a thin cannula through the rods of the bioreactor. The chamber was incubated at 37°C for 2h without flow and rotated every 30min. Medium flow started after 24h from seeding, at a speed of 0.5 ml/min.

3. If cellular colonization of scaffolds is a major hurdle, why was seeding not performed dynamically (under flow) using the bioreactor system?

Seeding of hMAB, mFB and mNCC directly into the lumen was not successful because, while cells could adhere, their penetration into the scaffold wall would not occur, even by increasing lumen pressure (data not shown). As explained in the methods, we have performed manual seeding using an insulin syringe with multiple injections into the muscle wall of the decellularized scaffolds, under a stereomicroscope. ROEC were seeded on the luminal surface of the oesophageal scaffolds and left in static culture conditions for 24h before starting the flow, if required.

The ‘seeding of cells’ section of the Methods has been improved for better clarity (lines 612-628).

4. The impact of cryo-preservation is not correctly assessed, as no quantification of cell death is performed (e.g. cas9, PI). Moreover, this was performed on a statically cultured scaffold -with a surprisingly limited cell distribution- while the dynamic condition is finally adopted. No evidences of a successful cryo-preservation of the final construct (after in vivo phase and ROEC seeding) are provided

Cryopreservation of scaffolds after seeding of muscle cells and static culture was shown as proof of principle. In order to assess the concern of the reviewer, we have increased the number of replicates, and performed MTT and immunohistochemistry for caspase3 in new samples cryopreserved for 2 weeks and subsequently cultured for up to 7 days (see figure below). Some of these results have been added to Figure 2 and to the Results section. Rare caspase3⁺ cells were found at 3 or 7 days of culture post-thawing (<1%). These data confirm the possibility of freezing and thawing seeded scaffolds for subsequent culture and seeding of ROEC.

No cryopreservation was attempted after *in vivo* implantation and ROEC seeding. The multi-stage approach described could be adopted for clinical translation with heterotopic transplantation of a muscle conduit allowing vascularization prior to delivery of the epithelial layer, followed by orthotopic transplantation to substitute the oesophagus. In this process, the new vasculature will be maintained during luminal seeding of epithelial cells and orthotopic transplantation. For this reason, cryopreservation post *in vivo* pre-vascularization and seeding of ROEC would not be within the scope of this work or approach.

Immunofluorescence staining for caspase3 and DAPI of a positive control and scaffolds cultured for 3 and 7 days after 2 weeks of cryopreservation. Arrows indicate cells positive for caspase3. Scale bar: 100 μ m.

5. Quantitative data should be systematically compiled when comparing static vs dynamic culture (e.g. Figure 4), in order to correctly assess spreading/engraftment/differentiation of mNCC in given conditions

We have added a quantification of GFP⁺, TuJ1⁺GFP⁺ and S100⁺GFP⁺ cells in static and dynamic conditions to Figure 4f. These results show that although the relative number of GFP⁺mNCC was significantly lower after dynamic culture, their level of differentiation into both neurons and glia was comparable between static and dynamic culture conditions. Moreover, during static culture mNCCs showed a very limited migratory capacity and tend to cluster at the injection sites, as shown in Fig.4b. The lower relative number of GFP+mNCC in dynamic culture conditions was eventually due to the higher growth of MABs in dynamic cultures.

The Results and Discussion have been changed accordingly.

6. The method is validated using 3 different cellular species (rat, mouse and human). The rationale behind should be clearly stated, and not solely justified by the ease of cellular identification within the grafts. This affects both clinical translation of the method, but also relevance of the presented data.

We thank the Reviewer for the interesting point. The use of different species facilitated their identification in the co-culture experiments and the analysis of cell fate after 3D culture. Use of different species in this novel 3D organ reconstruction has allowed us to exclude possible fusion and therefore be confident on the role of each cell type in tissue reconstitution. For a successful pre-clinical translation, all-human cells repopulated grafts will have to be developed and tested. Although we believe that cell-cell and cell-matrix interactions described in this study will promote comparable organ regeneration effects, testing all-human constructs will be essential to determine safety and functional outcomes for clinical translation.

These observations have been added to the Discussion (lines 433-441).

7. Neither the functionality nor the immunogenicity of the grafts is demonstrated.

Extent maturation of the oesophageal layers was shown both *in vitro* and *in vivo*, providing initial evidence of potential function. Mesoangioblasts were able to mature into smooth muscle cells, preserving a subset of progenitors with proliferative phenotype. Preliminary calcium imaging studies determined the functional status of mNCC suggesting that functional electrical connections were established within the scaffold providing evidence of a rudimentary circuitry. *In vivo* grafting of scaffolds in the omentum for 2 weeks showed mature and robust differentiation of muscle cells. *In vivo* functionality of the graft was not the scope of this study.

The inflammatory reaction to the construct *in vivo* was analysed with immunofluorescence of scaffolds 1 week post-implantation in the omentum of NSG mice. Results have been added to

Supplementary Figure 7g. A certain degree of expected inflammatory reaction was detected, with infiltration of F4/80+ macrophages and Ly6G+ neutrophils in the *muscularis externa* of the scaffold. This result is in line with published studies that highlight an initial inflammatory response to engineered grafts with decellularised scaffolds. The acute host response to ECM-derived scaffolds activates the innate immune response, including infiltration of neutrophils and macrophages. These scaffolds have been associated with a robust, but favourable host immune response that triggers constructive remodelling outcomes and an overall anti-inflammatory environment. In particular, macrophages have been recognized as a critical determinant of regeneration after injury, tissue remodelling and cross-talk with endogenous stem/progenitor cells [61,62]. These comments have been added to the Discussion (lines 394-403). We had previously shown the immunoregulatory effect of decellularized scaffolds in a discordant xenotransplantation model [Fishman et al. 2013]. *In vivo* preclinical studies will further provide essential information regarding the host immune response to the graft. Since we will test constructs obtained with all-human autologous cells, the immunogenicity will not be an issue for clinical translation. Although the study of the immunomodulatory effect of ECM-derived scaffolds is an interesting subject of investigation, it wasn't the scope of this study to address immunogenicity. This will be certainly explored in future pre-clinical experiments.

Fishman JM, Lowdell MW, Urbani L, Ansari T, Burns AJ, Turmaine M, North J, Sibbons P, Seifalian AM, Wood KJ, Birchall MA, De Coppi P. Immunomodulatory effect of a decellularized skeletal muscle scaffold in a discordant xenotransplantation model. PNAS USA. 2013 Aug 27;110(35):14360-5.

Specific comments:

- line 78: the statement “decellularization preserves architecture and composition” should be tempered, as no protein quantification is included. It could be valuable to quantitatively assess if the decellularization method preserves to some extent ECM proteins, in particular those capable to play a role in the subsequent guidance of differentiation

We thank the Reviewer for the comment. We have quantified the ECM components elastin and glycosaminoglycans to improve the characterization of the decellularized scaffold. The graphs are reported in Supplementary Figure 1f,g and show comparable amounts between native and decellularized oesophagi. Together with immunohistochemistry, we believe that these results show preservation of both architecture and composition of the native tissue, which are reflected in the mechanical properties of the scaffold. Collagen and sGAG are key ECM components of the 3D niche that guides cell differentiation. We have changed the Introduction and Results to temper the statement.

- Figure 1e: displaying triplicates is confusing. The authors should pick 1 representative scaffold per condition [ED: please move other examples to the supplement]

The choice of showing triplicates in Figure 1e was made to emphasize the difference in cellular distribution between scaffolds seeded with hMAB and hMAB+mFB. Schematic distribution maps are a visual representation of the cell distribution graph (Figure 1g) that shows high variability in hMAB-seeded scaffolds. We believe that, by selecting 1 representative map, the Figure would lose this important message, on which subsequent experiments are based on. The variability in distribution in hMAB-scaffolds would make it difficult to select 1 image, but we will be happy to move the other examples to a Supplementary Figure if the Reviewer and the Editor reckon that it will make the Figure clearer.

- Figure 2a: why not seeding also cells dynamically? If cells always seeded statically, dynamic culture does not increase engraftment, but rather proliferation of pre-seeded cells.

Part of this question has been already answered at Question 3: seeding of hMAB, mFB and mNCC directly into the lumen was not successful because, while cells could adhere, their penetration into the scaffold wall would not occur, even by increasing lumen pressure (data not shown). As explained in the methods, we have performed manual seeding using an insulin syringe with multiple injections into the muscle wall of the decellularized scaffolds, under a stereomicroscope. We have removed the term 'engraftment' from the Results as suggested by the Reviewer.

- Figure 3: it is not clear why some scaffolds were analysed after 6 days (figure 2) and others after 11 days. Does this correspond to 2 different phases of culture (e.g. 6 days of proliferation then differentiation till 11days)? Why not performing all analysis at the 11 days timepoint?

The results shown in Figure 1e-i were collected at 6 to 9 days of culture. In the set of experiments shown in Figure 1j,k, we focused on the migration of seeded cells and therefore we analysed an earlier time point (day 6). The main aim of this part of the study was to describe the positive effect of fibroblasts on the distribution and migration of mesoangioblasts, which could be masked by proliferation in prolonged cultures. Smooth muscle cell proliferation and differentiation were studied in the following set of experiments, which therefore required longer time points (day 11). Figure 2 and subsequent figures show analysis at 11 days of culture.

The Methods and Figure Legends now contain this information for better clarity (lines 631-632).

- Figure 3b: a comparative staining of native oesophageal muscle would be helpful.

It was not possible to provide a staining for the native oesophageal smooth muscle because the muscle layer of the rat oesophagus is entirely made of skeletal muscle fibres. The smooth muscle layer of a human oesophagus was used as comparison (Figure 3f). Distribution of cells in a native oesophageal muscle was also reported for comparison in Supplementary Figure 5a.

- Figure 5j: it is not clear if cells were dynamically seeded, or seeded and then cultured in bioreactor.

We apologise for the confusion. ROEC were seeded on the luminal surface of the oesophageal scaffold and left in static culture conditions for 24h before starting the flow (dynamic conditions). The 'seeding of cells' and 'static and dynamic culture' sections of the Methods have been detailed for better clarity (lines 612-656).

- Figure 6: there is no comparison (even semi-quantitative) of epithelium maturation between the in vivo extracted and the in vitro decellularized scaffold from figure 5.

We have conducted a qualitative analysis to test whether the oesophageal epithelial progenitor cells would be able to homogeneously cover a large surface (scaffold) and stratify while maintaining a basal proliferative layer important for regeneration and repair. The detection of a proliferative basal layer and of suprabasal layers expressing CK13 in both the *in vitro* and *in vivo* conditions, support the ability of the cultivated epithelial cells to generate a healthy, polarised and maturing esophageal epithelium. Direct comparison and quantification of *in vitro* and *in vivo* conditions would not be informative as the dynamic of differentiation and stratification *in vitro* and *in vivo* depends on very different factors (media and culture conditions in the first case and nutrients delivery *in vivo* in the second one).

Reviewer #2, expert in tissue engineering inc. oesophagus (Remarks to the Author):

Currently, there are two approaches to esophageal tissue engineering. One, the one undertaken by the authors, consists in constructing in vitro a substitute whose characteristics are as close as possible to the organ to be replaced. The other involves bringing into contact and implanting in vivo the elements considered necessary (extracellular matrix seeded with stem cells for example) so that tissue remodelling towards an esophageal phenotype occurs in vivo.

In the present work, 4 cell types are used for the making of the substitute, including 3 in co-culture (mesoangioblasts, fibroblasts and enteric neural cells). This work demonstrates that the co-culture of the first two cell types improves the distribution, homogeneity and migration of mesoangioblasts in the matrix; that cell colonization in the matrix is improved by dynamic incubation vs static incubation; that dynamic culture improves the differentiation of mesoangioblasts into smooth muscle cells; that it is possible to co-culture different cell types from different species; and finally that the substitute obtained by this approach has a structure close to that of the native esophagus, in vitro and after implantation in vivo. The approach is original because such a combination of cell types has never been tested in this context.

This work effectively analyzes the optimal conditions of construction of a finalized hybrid substitute. The amount of work is remarkable and the scientific demonstration rigorous, making this work convincing. No further evidence should be required to strengthen the conclusions of this paper. The level of detail from the experiments provided is adequate. However, one can question the possibility of application in human clinic of such an approach, given its complexity (4 cell types, sequential cell seeding). The question to which the authors have the project to answer and which underlies the merits of this approach is quid of the cellular arrangement and the survival of these cell populations after circumferential replacement of the esophagus by this substitute.

We thank the Reviewer for her/his interest in our work.

Regarding the cellular arrangement we have planned the approach considering from the beginning the potential of clinical translation and much of the work has been undertaken in partnership with the Centre for Cell, Gene & Tissue Therapeutics at UCL, which has successfully translated two cell/scaffold products to GMP compliance and over 20 other somatic cell therapies. To this aim, considering the possible application to patients with oesophageal atresia, human cells were isolated from tissues, which will normally be approached during clinical standard procedures to those patients. Mesangioblasts and fibroblasts have been isolated from skeletal muscle biopsies, which can be provided from the abdominal muscle wall of those patients at birth in case of a gastrostomy placement. These biopsies have been retrieved aseptically and subsequently processed such that they can be translated directly into a GMP-compliant manufacturing process. During the same procedure, neural crest cells (isolated from gut biopsies in this study) can be derived directly from the stomach. Finally the oesophageal epithelium can be derived from an endoscopic biopsy of the oesophagus during the initial assessment of the oesophagus at birth.

We fully accept the challenges which remain to complete the *ex-vivo* production of this oesophageal implant but we are confident that our experience with airways epithelial culture makes this feasible. Moreover, the translation to first-in-man and then routine production has been greatly facilitated by the design and manufacture of a bespoke “closed-system” bioreactor for decellularization and then recellularization of the donor oesophageal scaffold. The design of GMP compliant and transferable

devices, alongside the development of the product, is part of our unique approach to translational research.

Questions:

- is it not deleterious to differentiate mesoangioblasts before their implantation in vivo ? Is it not indeed preferable to preserve their ability to differentiate into other cell types and keep their capacity of secretions of factors involved in tissue regeneration, before implantation of the substitute in vivo ?

We thank the Reviewer for the interesting point. We agree that transplanting undifferentiated progenitors rather than fully differentiated cells usually results in higher engraftment. However, mesoangioblasts showed different levels of smooth muscle differentiation, with some cells not yet fully differentiated (calponin⁺ and connexin43⁺) and some still proliferating after 11 days of dynamic culture (about 13% of cells are Ki67⁺). This indicates that a subset of hMAB maintained a progenitor state. The organized differentiation towards smooth muscle observed after *in vivo* implantation highlighted that hMAB were able to express a differentiated phenotype also in the *in vivo* environment after dynamic culture. On a different note, another important message of our study is that the ECM of decellularized scaffolds and dynamic culture conditions guide and improve cell differentiation *in vitro*. These findings could have significant value in the field of tissue engineering for 3D culture models.

- what quality controls, apart from the structural analyzes, were carried out on the decellularized matrix (toxicity, immunogenicity, preservation of proteoglycans?).

Decellularized scaffolds were obtained with a detergent-enzymatic treatment (DET). The decellularization of oesophagi with this process has been already described by the authors for other animal models [5,29]. In these studies, scaffolds showed efficient removal of cellular content with preservation of ECM architecture and main components, reflecting in biomechanical properties comparable to native oesophagi. The data in this manuscript confirm that decellularized rat oesophageal scaffolds can be obtained with the same method. Furthermore, scaffolds obtained from other tissues with the same decellularization protocol have been shown to support the culture of tissue-specific cells, with no detectable cytotoxicity caused by the scaffold [28]. Our cell-seeding experiments confirmed that DET-derived scaffolds support cell proliferation, migration and differentiation without evidence of cytotoxicity. Quality controls were carried out routinely on the scaffolds to test the DNA content and presence of mycoplasma after sterilization with γ -irradiation. Quantification of glycosaminoglycans and elastin in native and decellularized oesophagi has been added to Supplementary Figure 1f,g and to the Results, showing preservation of these ECM components after decellularization.

The immunogenicity of decellularized scaffolds is an important feature of these biomaterials. The authors have already shown the immunoregulatory effect of decellularized scaffolds in a discordant xenotransplantation model [Fishman et al. 2013]. Although the study of the immunomodulatory effect of ECM-derived scaffolds is an interesting subject of investigation, it wasn't the scope of this study to address immunogenicity.

Fishman JM, Lowdell MW, Urbani L, Ansari T, Burns AJ, Turmaine M, North J, Sibbons P, Seifalian AM, Wood KJ, Birchall MA, De Coppi P. Immunomodulatory effect of a decellularized skeletal muscle scaffold in a discordant xenotransplantation model. PNAS USA. 2013 Aug 27;110(35):14360-5.

- the future perspectives in terms of tissue engineering, as the "intelligent matrix ", that contain factors implicated in tissue regeneration, without cell seeding, or other approach using cellularized matrix (see above). should be discuss

Thank you for the relevant comment. We have added the following points to the discussion (lines 298-306):

Avoiding cadaveric derived scaffolds (either of human or animal origin) and using synthetic polymers would have the advantage of having an off-shelf product and eliminate the potential risks of infections and organ shortage [32]. The synthetic scaffolds could be designed to recapitulate the various oesophageal layers and preloaded with specific growth factor capable to both allow cell proliferation and differentiation. This smart manufacturing, which may even avoid the need for cell seeding prior to transplantation making it cheaper and safer for patients is, however, still limited to small scale [33]. Avoiding cell seeding may also be possible in case the polymer is used as a stent to drive oesophageal regeneration but it is ultimately removed endoscopically [34], or for the repair of small oesophageal defects which do not affect the all oesophageal circumference [35].

- what is the interest of cryopreservation of the substitute after cell implantation, when it is supposed that cellularized substitutes for clinical application will certainly use autologous cells.

Thank you for giving the opportunity to clarify this point. In our previous clinical experience of tracheal transplantation timing has been crucial. Patients who may benefit of an engineered organ can be somehow predicted in advance of the necessary treatment and can be identified based on the failure of conventional treatments. However, when decided to adopt the tissue engineering solution, we had to wait for cell biopsy, expansion, scaffold decellularization and engineering of the entire construct. This can be particularly relevant on a more complex organ such as the oesophagus where multiple cell types are necessary. Having the possibility of cryopreserving an engineered organ give the possibility of considering the treatment in a shorter time helping solving a more urgent situation. Moreover, it resolves the problem of transporting the engineered organs. At the moment, transport is difficult with limitations related to distance (because of the time allowed outside the incubator), risks related to cell viability and contaminations. Having a cryopreserved organ increased the possibility of commercialization and benefit to a larger number of patients. We have clarified this in the final manuscript and emphasised those points in the discussion (lines 407-419).

- what is the raison for implantation of the substitute under the renal capsule. Are there differences in the maturation of the substitute between the two sites (large omentum and renal capsule)?

The two models (implantation in the omentum or underneath the kidney capsule) were compared for *in vivo* survival of cultured scaffolds and their neo-vascularization. The aim of this comparison was to determine if the graft vascularization timing in the two *in vivo* models was different and how the *in vivo* environment affects viability, organisation and differentiation of cells within the scaffolds. Of relevance, in the kidney capsule only a small size scaffold could be implanted, while a larger, stented scaffold could be easily inserted within the omentum. The omental implantation model was the one reported in the main Figure 6 for the positive results obtained in the neo-vascularization of the graft, preservation of muscle cell orientation and maturation and its suitability for clinical translation as pre-vascularization step before orthotopic implantation.

The Results section has been expanded for better clarity (lines 247-251).

- Is there an inflammatory reaction at the implantation site?

See also point 7 of reviewer 1. To address this point, we have performed immunofluorescence analysis of scaffolds 1 week post-implantation in the omentum of NSG mice. Results have been added to Supplementary Figure 7g and to the Results section. A certain degree of inflammatory reaction was detected, with infiltration of F4/80+ macrophages and Ly6G+ neutrophils in the *muscularis externa* of the scaffold. This result is in line with published studies that highlight an initial inflammatory response to engineered grafts with decellularised scaffolds. The acute host response to ECM-derived scaffolds activates the innate immune response, including infiltration of neutrophils and macrophages. These scaffolds have been associated with an immune modulation that triggers constructive remodelling outcomes and an overall anti-inflammatory environment. In particular, macrophages have been recognized as a critical determinant of regeneration after injury, tissue remodelling and cross-talk with endogenous stem/progenitor cells [61,62]. These comments have been added to the Discussion (lines 394-403). The authors have previously shown the immunoregulatory effect of decellularized scaffolds in a discordant xenotransplantation model [Fishman et al. 2013]. *In vivo* preclinical studies will further provide essential information regarding the host immune response to the graft. Since we will test constructs obtained with all-human autologous cells, the immunogenicity will not be an issue for clinical translation. Although the study of the immunomodulatory effect of ECM-derived scaffolds is an interesting subject of investigation, it wasn't the scope of this study to address immunogenicity. This will be certainly explored in future pre-clinical experiments.

Fishman JM, Lowdell MW, Urbani L, Ansari T, Burns AJ, Turmaine M, North J, Sibbons P, Seifalian AM, Wood KJ, Birchall MA, De Coppi P. Immunomodulatory effect of a decellularized skeletal muscle scaffold in a discordant xenotransplantation model. PNAS USA. 2013 Aug 27;110(35):14360-5.

In conclusion, favorable opinion for publication subject to modification of the discussion according to the remarks above

Reviewer #3, expert in oesophagus bioengineering (Remarks to the Author):

In this manuscript by Urbani and Camilli et al, the authors used a full thickness esophagus decellularized using a perfusion method of detergent-enzymatic treatment (DET), and repopulated with smooth muscle progenitor cells (hMAB), mouse fibroblasts (mFB), and murine enteric neural cells (mNCC); and allowed to differentiate in a static well setting or in dynamic setting with flow (bioreactor culture) for 11d. The triple cell-seeded scaffolds were then placed in the mouse omentum for 1 week to induce endogenous angiogenesis; and finally explanted and seeded with rat epithelial cells for 2 weeks. The authors also showed the ability to cryopreserve a cell-seeded scaffold with maintenance of viability of the seeded cells. Such a technique would be of interest to the regenerative medicine field and eventual clinical translation. The authors were relatively rigorous in their characterization of the scaffold; and of their cell types with immunolabeling, flow cytometry, and functional testing to determine the cell phenotype pre- and postimplantation as well as live cell imaging to determine the infiltration of the cells in the scaffold. However, the methods for scaffold decellularization were not provided and the metrics used to evaluate the extent of decellularization were inconsistent with established guidelines. The authors showed mature differentiation into skeletal muscle that was innervated, did show evidence of angiogenesis, and a mature epithelium, hence showing proof of concept of a tissue engineered, multi-layered esophagus.

We thank the Reviewer for her/his interest in our work. We understand the Reviewer main point, and we followed her/his suggestion to clarify the method for decellularization and the characterization of the scaffold, as better discussed below.

Decellularization was performed with two cycles of detergent-enzymatic treatment (DET) according to established protocols [28,29,31]. Briefly, the oesophageal lumen was perfused with continuous fluid delivery (iPumps) at 1 ml/min. Each DET cycle was composed of deionised water at 4°C for 24h, 4% sodium deoxycholate (SDC; Sigma) at room temperature (RT) for 4h and 2000 Kunitz DNase-I (Sigma) in 1 M NaCl at RT for 3h. The process was repeated for two cycles.

The protocol has been expanded in the Methods to include these details (lines 476-485).

Characterization of decellularization efficiency was performed and described in Figure 1 and Supplementary Figure 1. We have also added quantification of elastin and sGAG to the analysis previously presented. The graphs are reported in Supplementary Figure 1f,g and show comparable amounts between native and decellularized oesophagi. We believe that DNA content, absence of nuclei, preservation of overall structure, organization and composition of the ECM, and analysis of the biomechanical properties confirmed that the decellularized scaffold obtained was consistent with established guidelines.

Major revisions in this manuscript are required before it would be acceptable for publication.

Major comments:

1. The authors present an interesting approach. However, the barriers to clinical translation of this approach are significant and should be discussed. For example, this study utilizes 4 separate cell types that would need to be cultured for approximately five weeks prior to use. The cost of c-GMP facilities to ensure authenticity and sterility of each of the cultures; and especially the step implanting and explanting from the omentum, finally re-seeding after a surgical procedure. It would be too labor-intensive and costprohibitive for clinical translation. The cGMP considerations should be addressed as a limitation of the proposed approach. Do you have reason to believe this could be translated? The clinical translation would require the engineered esophageal construct to be anastomosed to the native esophagus. Suture retention strength is not addressed and viability in vivo has yet to be established.

Regarding the cellular arrangement we have planned the approach considering from the beginning the potential of clinical translation and much of the work has been undertaken in partnership with the Centre for Cell, Gene & Tissue Therapeutics at UCL which has successfully translated two cell/scaffold products to GMP compliance and over 20 other somatic cell therapies. To this aim, considering the possible application to patients with oesophageal atresia, human cells were isolated from tissues, which will normally be approached during clinical standard procedures to those patients. Mesangioblasts and fibroblasts have been isolated from skeletal muscle biopsies, which can be provided from the abdominal muscle wall of those patients at birth in case of a gastrostomy placement. These biopsies have been retrieved aseptically and subsequently processed such that they can be translated directly into a GMP-compliant manufacturing process. During the same procedure, neural crest cells (isolated from gut biopsies in this study) can be derived directly from the stomach. Finally the oesophageal epithelium can be derived from an endoscopic biopsy of the oesophagus during the initial assessment of the oesophagus at birth.

We fully accept the challenges, which remain to complete the *ex-vivo* production of this oesophageal implant but we are confident that our experience with airways epithelial culture makes this feasible. Moreover, the translation to first-in-man and then routine production has been greatly facilitated by the design and manufacture of a bespoke “closed-system” bioreactor for decellularization and then recellularization of the donor oesophageal scaffold. The design of GMP compliant and transferable devices, alongside the development of the product, is part of our unique approach to translational research.

We have emphasised some of these points in the discussion (lines 420-433).

Regarding the suture retention, the reviewer has identified a critical aspect and challenge in oesophageal replacement. Our experience in airways tissue engineering has highlighted the importance of successful anastomosis but, equally, the importance of reducing the granuloma formation at the anastomosis sites.

We believe that the biomechanical compatibility of the engineered construct is essential for its clinical success and our manufacturing process indeed aims at delivering an implant with the same viscoelasticity of the native tissue.

2. Figure 4d,e □ Where were different mNCC markers used in the dynamic versus static cultures? Were the same markers tried for both conditions?

Yes, the same markers (GFP, TuJ1, S100 and SM22) were used to immunolabelling mNCC and differentiated hMAB in both static and dynamic-cultured scaffolds. Although showing different combinations of markers for both static (b,c) and dynamic (d-f) conditions, Figure 4 mainly aims at highlighting the role of the bioreactor in enhancing cell differentiation and distribution, without overloading the overall figure.

3. Change “immunogenicity” to “antigenicity” line 284 pg 9 – ECM bioscaffolds will induce an inflammatory response, but need to acknowledge the difference between M1-like “proinflammation” and M2-like “constructive remodeling, immunomodulatory” activation of the response. ECM bioscaffolds, when appropriately decellularized, have been shown to induce the cellular response, but acellular ECM scaffolds have been shown to produce a more regulatory, anti-inflammatory phenotype as opposed to a pro-inflammatory phenotype when cells are part of the construct. (See Brown et. al.)

Citation for Comment 3: Brown, B. N., Valentin, J. E., Stewart-Akers, A. M., McCabe, G. P. & Badylak, S. F. Macrophage phenotype and remodeling outcomes in response to biologic scaffolds with and without a cellular component. *Biomaterials* 30, 1482–1491 (2009).

We thank the Reviewer for comment and we have changed the sentence as suggested.

4. Many representative figures were included – at times it was difficult to follow because the static and dynamic figures were juxtaposed in different ways: sometimes static versus dynamic conditions were left to right, top to bottom or at angles to each other - but if there is a way to make it consistent from figure to figure, it would be easier to follow.

We thank the Reviewer for the suggestion. Where possible, we have changed the orientation of static and dynamic conditions to make the figures more consistent and easier to follow. Static and dynamic images are now generally left to right. Figure 4 and 5 have been changed to this orientation to be consistent with Figure 2 and 3.

5. Controls showed inconsistencies. Comparisons were made between static versus dynamic for cell distribution, proliferation, skeletal muscle distribution, etc but implantation into the omentum was dynamic versus unseeded control, when that control was not shown previously; and then epithelial cells were finally seeded on both dynamic and static constructs. Why did the authors choose dynamic versus unseeded control instead of static for the omental implantation?

The comparison between static and dynamic conditions was essential to describe the superior culture conditions provided by the dynamic system. This *in vitro* comparison was maintained throughout the study to show consistency and a direct evaluation of cell distribution, proliferation and differentiation. Since we identified that the dynamic condition produced a better recellularized oesophagus, static-cultured scaffolds were not transplanted *in vivo*. We believe that the *in vitro* section of the study supports the choice of the dynamic condition and we considered that using another group of animals to test *in vivo* maturation of static scaffolds was not necessary. Unseeded scaffolds were used as controls to understand if invasion of host cells and vascularization of the scaffold were different between seeded and unseeded scaffolds.

6. How were the n values derived for the different characterization experiments, was a power analysis performed? Please be consistent when stating biological and technical replicates for the cell-based assays, and fields of view that were quantified for immunolabeling for each n.

Unless stated otherwise, characterization experiments were performed on at least 3 separate biological replicates. The n value reported in the manuscript for each analysis/assay represents the number of biological replicates. The only exceptions to this are the graphs in Figure 2g, 2h, 3h, and 4f which were obtained analysing technical replicates.

Seeding and 3D culture in static and dynamic conditions were performed on 2 or more scaffolds in parallel per experiment. For analysis of cell engraftment, counting, migration, density and marker expression, in general, a minimum of 3 scaffolds per analysis were used. Each scaffold was also cut in serial sections and analysed for multiple tests/assays when needed. For cell counting (using immunofluorescence or polar diagram analysis) we randomly selected immunostaining images

(3÷14), which were analysed and a single data point taken as the average cell counts across the analysed images, unless stated otherwise. *In vivo* transplantation studies were performed on the minimum number of animals to test cell survival and specification and angiogenesis (omentum implantation: n=3 seeded scaffold group, n=2 unseeded scaffold group; kidney capsule transplantation: n=4 seeded scaffold group, n=2 unseeded scaffold group). Feasibility of the *in vivo* transplantation models had already been tested in previous studies. No statistical methods were used to predetermine sample size.

We have added some of these details to the Methods (lines 785-791), and it can also be found in the revised version of the Reporting Summary requested by the Journal and linked to the Manuscript.

Minor comments:

7. The authors' review of the literature is deficient. Significant studies by Nieponice have shown promising results in preclinical animal studies and in human clinical studies. In addition, the authors failed to note the cohort study in which acellular scaffold materials successfully replaced neoplastic esophageal mucosa in five human patients. Although some of these studies were not full thickness defect, the authors should acknowledge these studies so that a comparison of cell-seeded versus acellular scaffolds for given indications can be evaluated.

We apologise for the omission. The Discussion has been improved following the Reviewer's suggestions incorporating more information on results from preclinical and clinical studies. The following paragraph has been added to the Discussion (lines 306-313):

Orthotopic implantation of unseeded decellularized scaffolds leads to stenosis, strictures and lack of function with poor clinical outcome [17, 19, 36, 37], nevertheless the positive impact of ECM from decellularized tissues on oesophageal healing process have been reported in a canine model [38]. Promising results have been shown also when unseeded decellularized matrices were used for oesophageal reconstruction in humans. Full-thickness patch repairs and circumferential mucosal substitution have been attempted with notable outcomes [39, 40]. More recently, an unseeded biological and synthetic combined scaffold was successfully used for bridging a circumferential full-thickness defect in 1 patient [35].

8. How do you know the hMAB smooth muscle differentiation was triggered by TGFbeta (as stated in the text line 186 pg 6) when it was not a treatment in supplemental figure 4?

We thank the Reviewer for comment. The analysis of the oxidative metabolism in 2D cultures shown in Supplementary Figure 4 was conducted in presence of TGFβ1 for both hMAB and hMAB+mFB culture conditions. The graph demonstrates that there was no difference in oxidative metabolism in hMAB versus hMAB+mFB, showing that smooth muscle differentiation was triggered by TGFβ1 only. We have added the missing information to the figure legend, and the Results has also been amended.

9. It should be stated in the results section that there were mature neurons and glial cells that were making connections – this shouldn't be brought up for the first time in the discussion line 336 pg 10

We thank the Reviewer for the comment. The Results has been amended to include this finding (line 207).

10. Pg 3 line 75 – change “but” to “and”

We have made the change.

11. Pg 3 line 80 – “and successful” The introduction could use a revision, some awkward phrasing

We have revised and improved the Introduction.

12. Recommend changing “fresh” to “native” throughout manuscript to be more clear

‘Fresh’ has been changed to ‘native’ in the main text and in all the figures and legends.

13. Fig 3C (what is different between the left and right? Could use labels)

Left: dynamic-cultured scaffold, right: unseeded (decellularized scaffold). We have added labels to the figure.

14. Labels wrong for fig 3f, g, h. line 184 pg 6?

We apologies for the confusion. The results (page 6) are correct, we amended the figure legend with the correct information.

15. Awkward sentence line 184 pg 6 “revealing... which revealed”

The sentence has been amended.

16. Line 184 first sentence of paragraph should be re-written, had to re-read several times to understand.

The paragraph has been re-written to improve its clarity (lines 186-191).

17. Figure 4B – only shows static (although refers to dynamic too in line 195, pg 6)

The main text has been amended.

18. Figure 4f was confusing for me to tell which was what, could use labels

Labels have been added to Figure 4F for better clarity.

19. Not a sentence line 201 pg 6 “in addition...”

The sentence has been re-written (line 207).

20. Spell out NSG line 241 pg 7

The sentence has been amended.

REVIEWERS' COMMENTS:

Reviewer #1 (Remarks to the Author):

Reviewer #1, expert in bioreactors and 3D cell culture scaffolds (Remarks to the Author):

In this study Urbani et al. propose a method to engineer oesophagus grafts by combining a rat decellularized oesophageal scaffold with human mesoangioblasts (hMAB), mouse fibroblasts (mFB), murine neural crest cells (mNCC) and rat oesophageal epithelial cells (ROEC) through sequential in vitro and in vivo phases. A bioreactor is used to improve cell distribution and enhance cell differentiation.

The manuscript is overall well-written and the authors provide a proof-of-principle for the successful generation of tissue-engineered oesophagus. Each component of the study (i.e., oesophagus decellularization, re-population with multiple cell types, use of a bioreactor system, comparison with native tissue) represents in itself a rather incremental step forward in the field, of descriptive nature. What would be required for reaching a convincing level of relevance is the demonstration that combination of all such elements leads to a functional superiority of the resulting graft over existing strategies. The twostage recellularization process is original and innovative, but again no proof of the need for such procedure to improve therapeutic outcome is offered. Thus, at least some of the perspectives listed at the end of the discussion (e.g., orthotopic implantation in a large animal model, host immune response to the graft, ECM remodelling, long-term contribution of seeded cells to oesophageal regeneration) should be included in a revised manuscript.

We thank the Reviewer for her/his interest in our work and for the comments. Some of the topics listed have been added to the Discussion and explored in the comments below (point 6 and 7).

The authors provided additional data and discussion points helping in the understanding of the manuscript. Unfortunately, the major suggestions to enhance the quality of the study (orthotopic evaluation of the constructs, deeper characterization of construct maturation and role of cells, or analysis of the immune response -despite a rather superficial attempt) were not seriously considered. It feels like the manuscript falls a bit short for publication in Nature Communications, especially in light of recent articles (e.g. “Decellularized and matured esophageal scaffold for circumferential esophagus replacement: Proof of concept in a pig model”, Biomaterials, May 2018).

Additional major issues:

1. A certain number of controls is missing, in particular regarding stainings for differentiation assessment which should be systematically performed on both undifferentiated and differentiated cells (e.g. Figure 1c/d, Figure 5d/e/f)

We thank the Reviewer for the comment. Characterization of undifferentiated hMAB is shown in Figure 1c with immunofluorescence for specific mesoangioblast markers. Differentiation of hMAB towards smooth muscle after incubation with TGF β 1 is shown in Supplementary Figure 2b. A panel of immunofluorescence for smooth muscle markers on undifferentiated hMAB (without TGF β 1) has been added to Supplementary Figure 2b as control. The capacity of hMAB to fully differentiate into smooth muscle cells is shown by positivity for smoothelin with immunofluorescence after 2 weeks of exposure to TGF β 1. A representative image has been added to Supplementary Figure 2b.

Discrepancies in aSMA staining is observed between undifferentiated hMAB (Fig1c, aSMA+) and undifferentiated hMAB (supFig2b, aSMA-/low).

Immunofluorescence of fibroblasts, reported in Figure 1d, identified cells as positive for Vimentin and Tcf4. Mouse NCC is a mixed population of undifferentiated cells, glia and neurons, as previously shown by the authors [55]. The expression of SOX10, S100 and TuJ1 by mNCC in culture was reported to confirm their phenotype before seeding experiments (Figure 4a).

Immunofluorescence of undifferentiated ROEC has been added in Supplementary Figure 6 to show absence of CK13 at day 4 of culture. **Ok.**

2. The authors claim the use of a novel “customized” bioreactor. No details on the bioreactor structure/design/functionality/operational parameters are provided

Thank you for the useful suggestion. We have added a simplified schematic of the bioreactor as Figure 2a to show the different components of the system. A detailed explanation of design and operational parameters has been added to the Methods (lines 630-656):

For dynamic cultures, seeded scaffolds were sutured to glass rods with 3-0 silk sutures and these were placed in a custom dual glass chamber. This chamber allows physical separation between lumen and external surface of the scaffold (as shown in Fig. 2b). A schematic representation of the bioreactor and its components used for dynamic culture is now reported in Fig. 2a. The bioreactor was designed to allow medium flow inside the lumen whereas an inlet and outlet present in the external chamber allowed medium flow around the scaffold. Lumen and external flows were controlled with an Applikon® bioreactor, connected to a reservoir of medium. The external chamber was filled up with medium and connected to the Applikon bioreactor. Medium flow was activated 6h after seeding. Medium flow was 5 ml/min. Glass chambers and medium reservoirs were cultured in a humidified incubator at 37°C and 5% CO₂. Medium was changed entirely every 2-3 days. Scaffolds were cultured for 2 days in proliferation medium (PM) and 9 days in smooth muscle differentiation medium, with TGFβ1 added daily through the injection point of the bioreactor.

[...] For seeding in the bioreactor, cells were delivered within the lumen of the oesophageal scaffold with a thin cannula through the rods of the bioreactor. The chamber was incubated at 37°C for 2h without flow and rotated every 30min. Medium flow started after 24h from seeding, at a speed of 0.5 ml/min. **Ok.**

3. If cellular colonization of scaffolds is a major hurdle, why was seeding not performed dynamically (under flow) using the bioreactor system?

Seeding of hMAB, mFB and mNCC directly into the lumen was not successful because, while cells could adhere, their penetration into the scaffold wall would not occur, even by increasing lumen pressure (data not shown). As explained in the methods, we have performed manual seeding using an insulin syringe with multiple injections into the muscle wall of the decellularized scaffolds, under a stereomicroscope. ROEC were seeded on the luminal surface of the oesophageal scaffolds and left in static culture conditions for 24h before starting the flow, if required. **Ok.**

The ‘seeding of cells’ section of the Methods has been improved for better clarity (lines 612-628).

4. The impact of cryo-preservation is not correctly assessed, as no quantification of cell death is performed (e.g. cas9, PI). Moreover, this was performed on a statically cultured scaffold -with a surprisingly limited cell distribution- while the dynamic condition is finally adopted. No evidences of a successful cryo-preservation of the final construct (after in vivo phase and ROEC seeding) are provided

Cryopreservation of scaffolds after seeding of muscle cells and static culture was shown as proof of principle. In order to assess the concern of the reviewer, we have increased the number of replicates, and performed MTT and immunohistochemistry for caspase3 in new samples cryopreserved for 2 weeks and subsequently cultured for up to 7 days (see figure below). Some of these results have been

added to Figure 2 and to the Results section. Rare caspase3⁺ cells were found at 3 or 7 days of culture post-thawing (<1%). These data confirm the possibility of freezing and thawing seeded scaffolds for subsequent culture and seeding of ROEC. **Figure 2I clearly shows impact of cryo on radiance (2 fold decrease). Still no data on dynamically cultured scaffold.**

No cryopreservation was attempted after *in vivo* implantation and ROEC seeding. The multi-stage approach described could be adopted for clinical translation with heterotopic transplantation of a muscle conduit allowing vascularization prior to delivery of the epithelial layer, followed by orthotopic transplantation to substitute the oesophagus. In this process, the new vasculature will be maintained during luminal seeding of epithelial cells and orthotopic transplantation. For this reason, cryopreservation post *in vivo* pre-vascularization and seeding of ROEC would not be within the scope of this work or approach. **Ok.**

Immunofluorescence staining for caspase3 and DAPI of a positive control and scaffolds cultured for 3 and 7 days after 2 weeks of cryopreservation. Arrows indicate cells positive for caspase3. Scale bar: 100µm.

5. Quantitative data should be systematically compiled when comparing static vs dynamic culture (e.g. Figure 4), in order to correctly assess spreading/engraftment/differentiation of mNCC in given conditions

We have added a quantification of GFP⁺, TuJ1⁺GFP⁺ and S100⁺GFP⁺ cells in static and dynamic conditions to Figure 4f. These results show that although the relative number of GFP⁺mNCC was significantly lower after dynamic culture, their level of differentiation into both neurons and glia was comparable between static and dynamic culture conditions. Moreover, during static culture mNCCs showed a very limited migratory capacity and tend to cluster at the injection sites, as shown in Fig.4b. The lower relative number of GFP⁺mNCC in dynamic culture conditions was eventually due to the higher growth of MABs in dynamic cultures. **These data seem to compromise the claimed superiority of the dynamic culture, at least for both mNCC seeding and differentiation. If the authors associate this to a possible “higher growth of MABs”, it should be demonstrated. In Figure 4b,c,d,e,g, a staining comparison static versus dynamic with the exact same marker combination would be valuable.**

The Results and Discussion have been changed accordingly.

6. The method is validated using 3 different cellular species (rat, mouse and human). The rationale behind should be clearly stated, and not solely justified by the ease of cellular identification within the grafts. This affects both clinical translation of the method, but also relevance of the presented data.

We thank the Reviewer for the interesting point. The use of different species facilitated their identification in the co-culture experiments and the analysis of cell fate after 3D culture. Use of different species in this novel 3D organ reconstruction has allowed us to exclude possible fusion and therefore be confident on the role of each cell type in tissue reconstitution. **Ok.** For a successful pre-clinical translation, all-human cells repopulated grafts will have to be developed and tested. Although we believe that cell-cell and cell-matrix interactions described in this study will promote comparable

organ regeneration effects, testing all-human constructs will be essential to determine safety and functional outcomes for clinical translation. These observations have been added to the Discussion (lines 433-441).

It sounds like a high degree of similarity across species is expected. However, as stated by the authors later on, rat and human oesophagus display a different composition (the muscle layer of the rat oesophagus is entirely made of skeletal muscle fibres). Such organizational/functional differences weaken the strategy to combine human/mouse/rat cells in the study and are underestimated in the introduced discussion point.

7. Neither the functionality nor the immunogenicity of the grafts is demonstrated.

Extent maturation of the oesophageal layers was shown both *in vitro* and *in vivo*, providing initial evidence of potential function. Mesoangioblasts were able to mature into smooth muscle cells, preserving a subset of progenitors with proliferative phenotype. Preliminary calcium imaging studies determined the functional status of mNCC suggesting that functional electrical connections were established within the scaffold providing evidence of a rudimentary circuitry. *In vivo* grafting of scaffolds in the omentum for 2 weeks showed mature and robust differentiation of muscle cells. *In vivo* functionality of the graft was not the scope of this study.

The inflammatory reaction to the construct *in vivo* was analysed with immunofluorescence of scaffolds 1 week post-implantation in the omentum of NSG mice. Results have been added to Supplementary Figure 7g. A certain degree of expected inflammatory reaction was detected, with infiltration of F4/80+ macrophages and Ly6G+ neutrophils in the *muscularis externa* of the scaffold. This result is in line with published studies that highlight an initial inflammatory response to engineered grafts with decellularised scaffolds. Testing the inflammatory reaction in an immunodeficient animals is not of the highest relevance. The detection of immune cell infiltration is indeed to be expected, as for any implanted material. Without quantification nor control material, the sole presentation of staining pictures is not much informative on the immunogenicity of the engineered oesophagus.

The acute host response to ECM-derived scaffolds activates the innate immune response, including infiltration of neutrophils and macrophages. These scaffolds have been associated with a robust, but favourable host immune response that triggers constructive remodelling outcomes and an overall anti-inflammatory environment. In particular, macrophages have been recognized as a critical determinant of regeneration after injury, tissue remodelling and cross-talk with endogenous stem/progenitor cells [61,62]. These comments have been added to the Discussion (lines 394-403). We had previously shown the immunoregulatory effect of decellularized scaffolds in a discordant xenotransplantation model [Fishman et al. 2013]. *In vivo* preclinical studies will further provide essential information regarding the host immune response to the graft. Since we will test constructs obtained with all-human autologous cells, the immunogenicity will not be an issue for clinical translation. Although the study of the immunomodulatory effect of ECM-derived scaffolds is an interesting subject of investigation, it wasn't the scope of this study to address immunogenicity. This will be certainly explored in future pre-clinical experiments.

Fishman JM, Lowdell MW, Urbani L, Ansari T, Burns AJ, Turmaine M, North J, Sibbons P, Seifalian AM, Wood KJ, Birchall MA, De Coppi P. Immunomodulatory effect of a decellularized skeletal muscle scaffold in a discordant xenotransplantation model. PNAS USA. 2013 Aug 27;110(35):14360-5.

Specific comments:

- line 78: the statement “decellularization preserves architecture and composition” should be tempered, as no protein quantification is included. It could be valuable to quantitatively assess if the decellularization method preserves to some extent ECM proteins, in particular those capable to play a role in the subsequent guidance of differentiation

We thank the Reviewer for the comment. We have quantified the ECM components elastin and glycosaminoglycans to improve the characterization of the decellularized scaffold. The graphs are reported in Supplementary Figure 1f,g and show comparable amounts between native and decellularized oesophagi. Together with immunohistochemistry, we believe that these results show preservation of both architecture and composition of the native tissue, which are reflected in the mechanical properties of the scaffold. Collagen and sGAG are key ECM components of the 3D niche that guides cell differentiation. We have changed the Introduction and Results to temper the statement. **Ok.**

- Figure 1e: displaying triplicates is confusing. The authors should pick 1 representative scaffold per condition [ED: please move other examples to the supplement]

The choice of showing triplicates in Figure 1e was made to emphasize the difference in cellular distribution between scaffolds seeded with hMAB and hMAB+mFB. Schematic distribution maps are a visual representation of the cell distribution graph (Figure 1g) that shows high variability in hMAB-seeded scaffolds. We believe that, by selecting 1 representative map, the Figure would lose this important message, on which subsequent experiments are based on. The variability in distribution in hMAB-scaffolds would make it difficult to select 1 image, but we will be happy to move the other examples to a Supplementary Figure if the Reviewer and the Editor reckon that it will make the Figure clearer. **Ok.**

- Figure 2a: why not seeding also cells dynamically? If cells always seeded statically, dynamic culture does not increase engraftment, but rather proliferation of pre-seeded cells.

Part of this question has been already answered at Question 3: seeding of hMAB, mFB and mNCC directly into the lumen was not successful because, while cells could adhere, their penetration into the scaffold wall would not occur, even by increasing lumen pressure (data not shown). As explained in the methods, we have performed manual seeding using an insulin syringe with multiple injections into the muscle wall of the decellularized scaffolds, under a stereomicroscope. We have removed the term ‘engraftment’ from the Results as suggested by the Reviewer. **ok.**

- Figure 3: it is not clear why some scaffolds were analysed after 6 days (figure 2) and others after 11 days. Does this correspond to 2 different phases of culture (e.g. 6 days of proliferation then differentiation till 11days)? Why not performing all analysis at the 11 days timepoint?

The results shown in Figure 1e-i were collected at 6 to 9 days of culture. In the set of experiments shown in Figure 1j,k, we focused on the migration of seeded cells and therefore we analysed an earlier time point (day 6). The main aim of this part of the study was to describe the positive effect of fibroblasts on the distribution and migration of mesoangioblasts, which could be masked by proliferation in prolonged cultures. Smooth muscle cell proliferation and differentiation were studied in the following set of experiments, which therefore required longer time points (day 11). Figure 2 and subsequent figures show analysis at 11 days of culture.

The Methods and Figure Legends now contain this information for better clarity (lines 631-632). **Ok.**

- Figure 3b: a comparative staining of native oesophageal muscle would be helpful.

It was not possible to provide a staining for the native oesophageal smooth muscle because the muscle layer of the rat oesophagus is entirely made of skeletal muscle fibres. The smooth muscle layer of a human oesophagus was used as comparison (Figure 3f). Distribution of cells in a native oesophageal muscle was also reported for comparison in Supplementary Figure 5a. **Ok.**

- Figure 5j: it is not clear if cells were dynamically seeded, or seeded and then cultured in bioreactor.

We apologise for the confusion. ROEC were seeded on the luminal surface of the oesophageal scaffold and left in static culture conditions for 24h before starting the flow (dynamic conditions). The ‘seeding of cells’ and ‘static and dynamic culture’ sections of the Methods have been detailed for better clarity (lines 612-656). **Ok.**

- Figure 6: there is no comparison (even semi-quantitative) of epithelium maturation between the in vivo extracted and the in vitro decellularized scaffold from figure 5.

We have conducted a qualitative analysis to test whether the oesophageal epithelial progenitor cells would be able to homogeneously cover a large surface (scaffold) and stratify while maintaining a basal proliferative layer important for regeneration and repair. The detection of a proliferative basal layer and of suprabasal layers expressing CK13 in both the *in vitro* and *in vivo* conditions, support the ability of the cultivated epithelial cells to generate a healthy, polarised and maturing esophageal epithelium. Direct comparison and quantification of *in vitro* and *in vivo* conditions would not be informative as the dynamic of differentiation and stratification *in vitro* and *in vivo* depends on very different factors (media and culture conditions in the first case and nutrients delivery *in vivo* in the second one). **Ok.**

Reviewer #2 (Remarks to the Author):

The answers given by the authors to my comments are quite satisfactory. I have no other remark that could delay the acceptance of this manuscript if that is the decision of the editor

Reviewer #3 (Remarks to the Author):

The authors have adequately addressed all of the concerns of the reviewers. Specifically, the expanded results and discussion sections as well as clarifications throughout the text strengthen the manuscript and improve the ease of reading.

Reviewer #1, expert in bioreactors and 3D cell culture scaffolds (Remarks to the Author):

In this study Urbani et al. propose a method to engineer oesophagus grafts by combining a rat decellularized oesophageal scaffold with human mesoangioblasts (hMAB), mouse fibroblasts (mFB), murine neural crest cells (mNCC) and rat oesophageal epithelial cells (ROEC) through sequential in vitro and in vivo phases. A bioreactor is used to improve cell distribution and enhance cell differentiation.

The manuscript is overall well-written and the authors provide a proof-of-principle for the successful generation of tissue-engineered oesophagus. Each component of the study (i.e., oesophagus decellularization, re-population with multiple cell types, use of a bioreactor system, comparison with native tissue) represents in itself a rather incremental step forward in the field, of descriptive nature. What would be required for reaching a convincing level of relevance is the demonstration that combination of all such elements leads to a functional superiority of the resulting graft over existing strategies. The twostage recellularization process is original and innovative, but again no proof of the need for such procedure to improve therapeutic outcome is offered. Thus, at least some of the perspectives listed at the end of the discussion (e.g., orthotopic implantation in a large animal model, host immune response to the graft, ECM remodelling, long-term contribution of seeded cells to oesophageal regeneration) should be included in a revised manuscript.

We thank the Reviewer for her/his interest in our work and for the comments. Some of the topics listed have been added to the Discussion and explored in the comments below (point 6 and 7).

The authors provided additional data and discussion points helping in the understanding of the manuscript. Unfortunately, the major suggestions to enhance the quality of the study (orthotopic evaluation of the constructs, deeper characterization of construct maturation and role of cells, or analysis of the immune response -despite a rather superficial attempt) were not seriously considered. It feels like the manuscript falls a bit short for publication in Nature Communications, especially in light of recent articles (e.g. “Decellularized and matured esophageal scaffold for circumferential esophagus replacement: Proof of concept in a pig model”, Biomaterials, May 2018).

We believe that extent maturation of the oesophageal layers was shown both *in vitro* and *in vivo*, providing initial evidence of potential function. This study was a proof of principle to validate the engineering technique and obtain a multi-layered tissue with structural and cellular organization that resemble a native oesophagus. As also clarified in the comments below, functionality and long-term contribution of seeded cells to oesophageal regeneration will be explored in vivo future experiments with all human-seeded scaffolds.

Regarding the recent publication in Biomaterials highlighted by the reviewer, we believe our paper still contributes significantly to the field. Various points distinguish our work as discussed in the amended manuscript. In particular, the authors use a mixture of adipose derived cells to seed the decellularised oesophagus before implantation, which might help and not drive in vivo regeneration. On the contrary, we show that tissue-specific stem/progenitor cells with the potency of differentiating both ex vivo and in vivo lead to the regeneration of a functional oesophagus.

Regarding the immune response to the graft, please see point 7 below.

The aforementioned research article on Biomaterials has been included in the Discussion (page 9, 11 and 13).

Additional major issues:

1. A certain number of controls is missing, in particular regarding stainings for differentiation assessment which should be systematically performed on both undifferentiated and differentiated cells (e.g. Figure 1c/d, Figure 5d/e/f)

We thank the Reviewer for the comment. Characterization of undifferentiated hMAB is shown in Figure 1c with immunofluorescence for specific mesoangioblast markers. Differentiation of hMAB towards smooth muscle after incubation with TGF β 1 is shown in Supplementary Figure 2b. A panel of immunofluorescence for smooth muscle markers on undifferentiated hMAB (without TGF β 1) has been added to Supplementary Figure 2b as control. The capacity of hMAB to fully differentiate into smooth muscle cells is shown by positivity for smoothelin with immunofluorescence after 2 weeks of exposure to TGF β 1. A representative image has been added to Supplementary Figure 2b.

Discrepancies in α SMA staining is observed between undifferentiated hMAB (Fig1c, α SMA+) and undifferentiated hMAB (supFig2b, α SMA-/low).

The pattern of α SMA expression is different between the two figures because images were taken with different microscopes at different times and with different exposure. Exposure was determined in respect to appropriate internal positive and negative controls for the staining.

Immunofluorescence of fibroblasts, reported in Figure 1d, identified cells as positive for Vimentin and Tcf4. Mouse NCC is a mixed population of undifferentiated cells, glia and neurons, as previously shown by the authors [55]. The expression of SOX10, S100 and TuJ1 by mNCC in culture was reported to confirm their phenotype before seeding experiments (Figure 4a).

Immunofluorescence of undifferentiated ROEC has been added in Supplementary Figure 6 to show absence of CK13 at day 4 of culture. **Ok.**

2. The authors claim the use of a novel “customized” bioreactor. No details on the bioreactor structure/design/functionality/operational parameters are provided

Thank you for the useful suggestion. We have added a simplified schematic of the bioreactor as Figure 2a to show the different components of the system. A detailed explanation of design and operational parameters has been added to the Methods (lines 630-656):

For dynamic cultures, seeded scaffolds were sutured to glass rods with 3-0 silk sutures and these were placed in a custom dual glass chamber. This chamber allows physical separation between lumen and external surface of the scaffold (as shown in Fig. 2b). A schematic representation of the bioreactor and its components used for dynamic culture is now reported in Fig. 2a. The bioreactor was designed to allow medium flow inside the lumen whereas an inlet and outlet present in the external chamber allowed medium flow around the scaffold. Lumen and external flows were controlled with an Applikon® bioreactor, connected to a reservoir of medium. The external chamber was filled up with medium and connected to the Applikon bioreactor. Medium flow was activated 6h after seeding.

Medium flow was 5 ml/min. Glass chambers and medium reservoirs were cultured in a humidified incubator at 37°C and 5% CO₂. Medium was changed entirely every 2-3 days. Scaffolds were cultured for 2 days in proliferation medium (PM) and 9 days in smooth muscle differentiation medium, with TGF β 1 added daily through the injection point of the bioreactor.

[...] For seeding in the bioreactor, cells were delivered within the lumen of the oesophageal scaffold with a thin cannula through the rods of the bioreactor. The chamber was incubated at 37°C for 2h without flow and rotated every 30min. Medium flow started after 24h from seeding, at a speed of 0.5 ml/min. **Ok.**

3. If cellular colonization of scaffolds is a major hurdle, why was seeding not performed dynamically (under flow) using the bioreactor system?

Seeding of hMAB, mFB and mNCC directly into the lumen was not successful because, while cells could adhere, their penetration into the scaffold wall would not occur, even by increasing lumen pressure (data not shown). As explained in the methods, we have performed manual seeding using an insulin syringe with multiple injections into the muscle wall of the decellularized scaffolds, under a stereomicroscope. ROEC were seeded on the luminal surface of the oesophageal scaffolds and left in static culture conditions for 24h before starting the flow, if required. **Ok.**

The 'seeding of cells' section of the Methods has been improved for better clarity (lines 612-628).

4. The impact of cryo-preservation is not correctly assessed, as no quantification of cell death is performed (e.g. cas9, PI). Moreover, this was performed on a statically cultured scaffold -with a surprisingly limited cell distribution- while the dynamic condition is finally adopted. No evidences of a successful cryo-preservation of the final construct (after in vivo phase and ROEC seeding) are provided

Cryopreservation of scaffolds after seeding of muscle cells and static culture was shown as proof of principle. In order to assess the concern of the reviewer, we have increased the number of replicates, and performed MTT and immunohistochemistry for caspase3 in new samples cryopreserved for 2 weeks and subsequently cultured for up to 7 days (see figure below). Some of these results have been added to Figure 2 and to the Results section. Rare caspase3⁺ cells were found at 3 or 7 days of culture post-thawing (<1%). These data confirm the possibility of freezing and thawing seeded scaffolds for subsequent culture and seeding of ROEC. **Figure 2I clearly shows impact of cryo on radiance (2 fold decrease). Still no data on dynamically cultured scaffold.**

Post-thawing, scaffolds showed a slight reduction in cell viability when compared to before cryopreservation. However, cells were able to recover and grow for up to 7 days in static culture. This indicated that cryopreservation had an impact on cell viability but they were able to recover, also supported by histology and the presence of rare caspase3⁺ cells.

Cryopreservation of scaffolds after seeding of muscle cells and static culture was shown as proof of principle. Cryopreservation of dynamic-cultured scaffolds would need further optimization.

No cryopreservation was attempted after *in vivo* implantation and ROEC seeding. The multi-stage approach described could be adopted for clinical translation with heterotopic transplantation of a muscle conduit allowing vascularization prior to delivery of the epithelial layer, followed by orthotopic transplantation to substitute the oesophagus. In this process, the new vasculature will be maintained during luminal seeding of epithelial cells and orthotopic transplantation. For this reason, cryopreservation post *in vivo* pre-vascularization and seeding of ROEC would not be within the scope of this work or approach. **Ok.**

Immunofluorescence staining for caspase3 and DAPI of a positive control and scaffolds cultured for 3 and 7 days after 2 weeks of cryopreservation. Arrows indicate cells positive for caspase3. Scale bar: 100µm.

5. Quantitative data should be systematically compiled when comparing static vs dynamic culture (e.g. Figure 4), in order to correctly assess spreading/engraftment/differentiation of mNCC in given conditions

We have added a quantification of GFP⁺, TuJ1⁺GFP⁺ and S100⁺GFP⁺ cells in static and dynamic conditions to Figure 4f. These results show that although the relative number of GFP+mNCC was significantly lower after dynamic culture, their level of differentiation into both neurons and glia was comparable between static and dynamic culture conditions. Moreover, during static culture mNCCs showed a very limited migratory capacity and tend to cluster at the injection sites, as shown in Fig.4b. The lower relative number of GFP+mNCC in dynamic culture conditions was eventually due to the higher growth of MABs in dynamic cultures. **These data seem to compromise the claimed superiority of the dynamic culture, at least for both mNCC seeding and differentiation. If the authors associate this to a possible “higher growth of MABs”, it should be demonstrated. In Figure 4b,c,d,e,g, a staining comparison static versus dynamic with the exact same marker combination would be valuable.**

We think that the superiority of the dynamic culture is not compromised by these data. The higher growth of hMAB was demonstrated with qualitative and quantitative analysis as reported in Figure 2. Since the number of GFP⁺ cells was determined as a percentage of the total number of DAPI⁺ cells, we suggested that the significant lower percentage of GFP⁺ cells in dynamic culture could be related to a relative increase of the number of hMAB. In static culture, mNCC were mainly organized in clusters, while the dynamic culture induced their spreading from the injection site and producing in ring-like structures similar to those observed in native oesophagi – which might represent an advantage for functional integration in vivo.

Figure 4 and relative Legend were improved following the Reviewer' suggestion.

The Results and Discussion have been changed accordingly.

6. The method is validated using 3 different cellular species (rat, mouse and human). The rationale behind should be clearly stated, and not solely justified by the ease of cellular identification within the grafts. This affects both clinical translation of the method, but also relevance of the presented data.

We thank the Reviewer for the interesting point. The use of different species facilitated their identification in the co-culture experiments and the analysis of cell fate after 3D culture. Use of different species in this novel 3D organ reconstruction has allowed us to exclude possible fusion and therefore be confident on the role of each cell type in tissue reconstitution. **Ok.** For a successful preclinical translation, all-human cells repopulated grafts will have to be developed and tested. Although we believe that cell-cell and cell-matrix interactions described in this study will promote comparable organ regeneration effects, testing all-human constructs will be essential to determine safety and functional outcomes for clinical translation. These observations have been added to the Discussion (lines 433-441).

It sounds like a high degree of similarity across species is expected. However, as stated by the authors later on, rat and human oesophagus display a different composition (the muscle layer of the rat oesophagus is entirely made of skeletal muscle fibres). Such organizational/functional differences weaken the strategy to combine human/mouse/rat cells in the study and are underestimated in the introduced discussion point.

The different composition in muscle fibres between human and rat could potentially affect cell behaviour in the scaffold. However, for the muscle layers reconstitution we used human cells and the data demonstrate that the differentiation potency towards smooth muscle is achieved despite the scaffold originated from rat. As previously stated, the use of different species facilitated their identification in co-culture experiments also to exclude possible fusion. We recognize the limitations of this study and suggested that for a successful preclinical translation, all-human cells repopulated

grafts will have to be developed and tested to determine safety and functional outcomes. Following the Reviewer comment, the potential effect of skeletal versus smooth muscle-derived scaffolds on cells has been discussed in the Manuscript (page 12):

‘Decellularized oesophageal scaffolds alternative from rat origin will have different composition in muscle fibres since rat oesophagi contain only skeletal muscle. The potential effect of this change in muscle layer composition in cell behaviour in the scaffold should also be considered in future investigations.’

7. Neither the functionality nor the immunogenicity of the grafts is demonstrated.

Extent maturation of the oesophageal layers was shown both *in vitro* and *in vivo*, providing initial evidence of potential function. Mesoangioblasts were able to mature into smooth muscle cells, preserving a subset of progenitors with proliferative phenotype. Preliminary calcium imaging studies determined the functional status of mNCC suggesting that functional electrical connections were established within the scaffold providing evidence of a rudimentary circuitry. *In vivo* grafting of scaffolds in the omentum for 2 weeks showed mature and robust differentiation of muscle cells. *In vivo* functionality of the graft was not the scope of this study.

The inflammatory reaction to the construct *in vivo* was analysed with immunofluorescence of scaffolds 1 week post-implantation in the omentum of NSG mice. Results have been added to Supplementary Figure 7g. A certain degree of expected inflammatory reaction was detected, with infiltration of F4/80⁺ macrophages and Ly6G⁺ neutrophils in the *muscularis externa* of the scaffold. This result is in line with published studies that highlight an initial inflammatory response to engineered grafts with decellularised scaffolds. **Testing the inflammatory reaction in an immunodeficient animals is not of the highest relevance. The detection of immune cell infiltration is indeed to be expected, as for any implanted material. Without quantification nor control material, the sole presentation of staining pictures is not much informative on the immunogenicity of the engineered oesophagus.**

Since the oesophagus was engineered with human, rat and mouse cells, testing the inflammatory reaction in an immunocompetent animal model would have triggered rejection providing no valuable information. The immunogenicity of the decellularized scaffold has been already described by our and other groups, and was not the scope of this study. *In vivo* preclinical studies will further provide essential information regarding the host immune response to the graft, but they will be performed with all human cells. For clinical translation, we envisage the use of all human autologous cells, where the immunogenicity will not be an issue.

The acute host response to ECM-derived scaffolds activates the innate immune response, including infiltration of neutrophils and macrophages. These scaffolds have been associated with a robust, but favourable host immune response that triggers constructive remodelling outcomes and an overall anti-inflammatory environment. In particular, macrophages have been recognized as a critical determinant of regeneration after injury, tissue remodelling and cross-talk with endogenous stem/progenitor cells [61,62]. These comments have been added to the Discussion (lines 394-403). We had previously shown the immunoregulatory effect of decellularized scaffolds in a discordant xenotransplantation model [Fishman et al. 2013]. *In vivo* preclinical studies will further provide essential information regarding the host immune response to the graft. Since we will test constructs obtained with all human autologous cells, the immunogenicity will not be an issue for clinical translation. Although the study of the immunomodulatory effect of ECM-derived scaffolds is an interesting subject of investigation, it wasn't the scope of this study to address immunogenicity. This will be certainly explored in future pre-clinical experiments.

Fishman JM, Lowdell MW, Urbani L, Ansari T, Burns AJ, Turmaine M, North J, Sibbons P, Seifalian AM, Wood KJ, Birchall MA, De Coppi P. Immunomodulatory effect of a decellularized skeletal muscle scaffold in a discordant xenotransplantation model. PNAS USA. 2013 Aug 27;110(35):14360-5.

Specific comments:

- line 78: the statement “decellularization preserves architecture and composition” should be tempered, as no protein quantification is included. It could be valuable to quantitatively assess if the decellularization method preserves to some extent ECM proteins, in particular those capable to play a role in the subsequent guidance of differentiation

We thank the Reviewer for the comment. We have quantified the ECM components elastin and glycosaminoglycans to improve the characterization of the decellularized scaffold. The graphs are reported in Supplementary Figure 1f,g and show comparable amounts between native and decellularized oesophagi. Together with immunohistochemistry, we believe that these results show preservation of both architecture and composition of the native tissue, which are reflected in the mechanical properties of the scaffold. Collagen and sGAG are key ECM components of the 3D niche that guides cell differentiation. We have changed the Introduction and Results to temper the statement. **Ok.**

- Figure 1e: displaying triplicates is confusing. The authors should pick 1 representative scaffold per condition [ED: please move other examples to the supplement]

The choice of showing triplicates in Figure 1e was made to emphasize the difference in cellular distribution between scaffolds seeded with hMAB and hMAB+mFB. Schematic distribution maps are a visual representation of the cell distribution graph (Figure 1g) that shows high variability in hMABseeded scaffolds. We believe that, by selecting 1 representative map, the Figure would lose this important message, on which subsequent experiments are based on. The variability in distribution in hMAB-scaffolds would make it difficult to select 1 image, but we will be happy to move the other examples to a Supplementary Figure if the Reviewer and the Editor reckon that it will make the Figure clearer. **Ok.**

- Figure 2a: why not seeding also cells dynamically? If cells always seeded statically, dynamic culture does not increase engraftment, but rather proliferation of pre-seeded cells.

Part of this question has been already answered at Question 3: seeding of hMAB, mFB and mNCC directly into the lumen was not successful because, while cells could adhere, their penetration into the scaffold wall would not occur, even by increasing lumen pressure (data not shown). As explained in the methods, we have performed manual seeding using an insulin syringe with multiple injections into the muscle wall of the decellularized scaffolds, under a stereomicroscope. We have removed the term ‘engraftment’ from the Results as suggested by the Reviewer. **ok.**

- Figure 3: it is not clear why some scaffolds were analysed after 6 days (figure 2) and others after 11 days. Does this correspond to 2 different phases of culture (e.g. 6 days of proliferation then differentiation till 11days)? Why not performing all analysis at the 11 days timepoint?

The results shown in Figure 1e-i were collected at 6 to 9 days of culture. In the set of experiments shown in Figure 1j,k, we focused on the migration of seeded cells and therefore we analysed an earlier time point (day 6). The main aim of this part of the study was to describe the positive effect of fibroblasts on the distribution and migration of mesoangioblasts, which could be masked by proliferation in prolonged cultures. Smooth muscle cell proliferation and differentiation were studied

in the following set of experiments, which therefore required longer time points (day 11). Figure 2 and subsequent figures show analysis at 11 days of culture. The Methods and Figure Legends now contain this information for better clarity (lines 631-632). **Ok.**

- Figure 3b: a comparative staining of native oesophageal muscle would be helpful.

It was not possible to provide a staining for the native oesophageal smooth muscle because the muscle layer of the rat oesophagus is entirely made of skeletal muscle fibres. The smooth muscle layer of a human oesophagus was used as comparison (Figure 3f). Distribution of cells in a native oesophageal muscle was also reported for comparison in Supplementary Figure 5a. **Ok.**

- Figure 5j: it is not clear if cells were dynamically seeded, or seeded and then cultured in bioreactor.

We apologise for the confusion. ROEC were seeded on the luminal surface of the oesophageal scaffold and left in static culture conditions for 24h before starting the flow (dynamic conditions). The ‘seeding of cells’ and ‘static and dynamic culture’ sections of the Methods have been detailed for better clarity (lines 612-656). **Ok.**

- Figure 6: there is no comparison (even semi-quantitative) of epithelium maturation between the in vivo extracted and the in vitro decellularized scaffold from figure 5.

We have conducted a qualitative analysis to test whether the oesophageal epithelial progenitor cells would be able to homogeneously cover a large surface (scaffold) and stratify while maintaining a basal proliferative layer important for regeneration and repair. The detection of a proliferative basal layer and of suprabasal layers expressing CK13 in both the *in vitro* and *in vivo* conditions, support the ability of the cultivated epithelial cells to generate a healthy, polarised and maturing esophageal epithelium. Direct comparison and quantification of *in vitro* and *in vivo* conditions would not be informative as the dynamic of differentiation and stratification *in vitro* and *in vivo* depends on very different factors (media and culture conditions in the first case and nutrients delivery *in vivo* in the second one). **Ok.**